# Single-cell transcriptomic analysis suggests two molecularly distinct subtypes of intrahepatic cholangiocarcinoma

Guohe Song [1,10], Yang Shi[2,10], Lu Meng[3,10], Jiaqiang Ma[1,3,10], Siyuan Huang[4], Juan Zhang[1], Yingcheng Wu[1], Jiaxin Li[4], Youpei Lin[1], Shuaixi Yang[1], Dongning Rao[1], Yifei Cheng[1], Jian Lin[5], Shuyi Ji[5], Yuming Liu[1], Shan Jiang[3], Xiaoliang Wang[6], Shu Zhang[1], Aiwu Ke[1], Xiaoying Wang[1], Ya Cao [7], Yuan Ji [8], Jian Zhou [1,9], Jia Fan [1,9✉], Xiaoming Zhang [3✉], Ruibin Xi [2✉] & Qiang Gao [1✉]

Intrahepatic cholangiocarcinoma (iCCA) is a highly heterogeneous cancer with limited understanding of its classification and tumor microenvironment. Here, by performing single-cell RNA sequencing on 144,878 cells from 14 pairs of iCCA tumors and non-tumor liver tissues, we find that S100P and SPP1 are two markers for iCCA perihilar large duct type (iCCA[phl]) and peripheral small duct type (iCCA[pps]). S100P + SPP1− iCCA[phl] has significantly reduced levels of infiltrating CD4[+] T cells, CD56[+] NK cells, and increased CCL18[+] macrophages and PD1[+]CD8[+] T cells compared to S100P-SPP1 + iCCA[pps]. The transcription factor CREB3L1 is identified to regulate the S100P expression and promote tumor cell invasion. S100P-SPP1 + iCCA[pps] has significantly more SPP1[+] macrophage infiltration, less aggressiveness and better survival than S100P + SPP1− iCCA[phl]. Moreover, S100P-SPP1 + iCCA[pps] harbors tumor cells at different status of differentiation, such as ALB + hepatocyte differentiation and ID3+ stemness. Our study extends the understanding of the diversity of tumor cells in iCCA.

[1] Department of Liver Surgery and Transplantation, Liver Cancer Institute, Zhongshan Hospital, and Key Laboratory of Carcinogenesis and Cancer Invasion of Ministry of Education, Fudan University, Shanghai, China. [2] School of Mathematical Sciences and Center for Statistical Science, Peking University, Beijing, China. [3] Key Laboratory of Molecular Virology & Immunology, Institut Pasteur of Shanghai, Chinese Academy of Sciences, Shanghai, China. [4] Peking-Tsinghua Center for Life Sciences, Academy for Advanced Interdisciplinary Studies, Peking University, Beijing, China. [5] Department of Cancer Center, Jin Shan Hospital, Fudan University, Shanghai, China. [6] Department of General Surgery, Qingpu Branch of Zhongshan Hospital Affiliated to Fudan University, Shanghai, China. [7] Cancer Research Institute, Xiangya School of Medicine, Central South University, Changsha, Hunan, China. [8] Department of Pathology, Zhongshan Hospital, Fudan University, Shanghai, China. [9] Key Laboratory of Medical Epigenetics and Metabolism, Institutes of Biomedical Sciences, Fudan University, Shanghai, China. [10] These authors contributed equally: Guohe Song, Yang Shi, Lu Meng, Jiaqiang Ma. ✉email: fan.jia@zs-hospital.sh.cn; xmzhang@ips.ac.cn; ruibinxi@math.pku.edu.cn; gaoqiang@fudan.edu.cn

ntrahepatic cholangiocarcinoma (iCCA) is the second most common primary liver malignancy after hepatocellular carcinoma, with poor outcome and rising incidence globally[1]. As a highly heterogeneous disease, iCCA can originate from cholangiocytes located at any point of a biliary tree above the second-order bile ducts. Recently, the World Health Organization and European Network for the Study of Cholangiocarcinoma have recognized that iCCA can be classified into two main histologically distinct subtypes, including perihilar large duct type (iCCA$^{phl}$) and peripheral small duct type (iCCA$^{pps}$), according to the level or size of the affected bile duct[2,3]. Indeed, emerging evidence has indicated that the two histological subtypes of iCCA harbored distinct cellular origins and pathogenesis[4].

Generally, iCCA$^{phl}$ is considered to be derived from large intrahepatic bile ducts and mainly composed of mucin-producing cholangiocytes. This subtype of iCCA is characterized by mucus hypersecretion and has higher lymph node metastasis rates and worse survival[5] compared with iCCA$^{pps}$. It has been reported that MUC5AC, one of the main components of mucus, is frequently overexpressed in iCCA$^{phl}$ and associated with aggressive tumor behavior[6]. Also, S100P, a member of the S100 family of EF-hand calcium-binding proteins, that are highly expressed in various types of cancer and play crucial roles in tumor progression[7], is also upregulated in mucin-producing iCCAs and suggested to be an important marker[8,9] for iCCA$^{phl}$. On the contrary, iCCA$^{pps}$ is commonly believed to originate from small intrahepatic bile ducts with no or minimal mucin production. It has been found that iCCA$^{pps}$ express CDH2 more frequently than iCCA$^{phl}$ and present distinctive clinical and molecular features[10]. Moreover, NCAM, a marker of hepatic progenitor cells, was also expressed in iCCA$^{pps}$, as well as cholangiolocellular carcinoma (CLC) which is thought to originate from canals of Hering/bile ductules[3,5]. Although the two subtypes of iCCA displayed significant differences in mucin production, the shape of tumor cells, and patient prognosis[3,4], there is no consensus and definite panel of markers to distinguish them, and our knowledge of their biological, molecular, and therapeutic difference is still limited. Single-cell RNA sequencing (scRNA-seq) is a powerful technology for cancer research. Previous scRNA-seq studies have reported the complexity of the tumor microenvironment in iCCAs without taking into consideration of the histological classification, which may not accurately reflect the diversity of this tumor[11,12].

Here, we identify and independently validate that SPP1, together with S100P, are optimal discriminatory biomarkers for iCCA$^{phl}$ and iCCA$^{pps}$. As compared with S100P-SPP1+ iCCA$^{pps}$, S100P + SPP1− iCCA$^{phl}$ has increased CCL18+ macrophages infiltration, decreased SPP1+ macrophages, aggressive phenotypes, and worse prognosis. Our data further our understanding of the diversity of tumor cells in iCCA.

## Results

### Single-cell profiling of the tumor ecosystem in iCCA.
We applied scRNA-seq and whole-exome sequencing (WES) on tumor and paired adjacent non-tumor liver tissues from fourteen treatment-naïve iCCA patients (Fig. 1a and Supplementary Fig. 1a). All tumors were negative for Hep-Par1 and Arg-1 (specific markers for hepatocellular carcinoma) expression (Supplementary Fig. 1b). The patient clinicopathological characteristics are presented in Supplementary Data 1. We obtained single-cell transcriptomes for 144,878 cells after quality control. Thirteen main cell clusters with the expression of known marker genes were identified including epithelial cells, monocytes, macrophages, dendritic cells (DC), natural killer (NK) cells, CD4+ T cells, regulatory T cells (Treg), CD8+ T cells, mucosal-

associated invariant T (MAIT) cells, B cells, plasma B cells, fibroblasts, and endothelial cells (Fig. 1b). Totally, we identified 23,667 malignant cells by inferring large-scale copy number variations (CNVs) from epithelial cells with the high expression of KRT19 (Fig. 1c and Supplementary Fig. 1c). Consistent with previous findings in other tumors, malignant cells showed strong intertumoral heterogeneity and formed patient-specific clusters[13,14] (Fig. 1d). Also, infiltrating immune cells were found to be significantly heterogeneous among different patients and between tumor and peri-tumor tissues (Supplementary Fig. 1d, e). For example, macrophages, CD4+ T cells, and Tregs were highly infiltrated in the tumor, while MAIT cells were mainly distributed in the adjacent liver tissues (Fig. 1e).

### SPP1 is a representative marker for iCCA$^{pps}$.
To explore the subtypes of iCCA with different cell origins at the single-cell level, we examined the expression of previously proposed markers of iCCA$^{phl}$ (S100P, MUC5AC, and MUC6) and iCCA$^{pps}$ (NCAM1 and CDH2) in malignant cells[2,3] (Fig. 2a and Supplementary Fig. 2a). We found that 7 out of 14 iCCAs (P02, P03, P04, P06, P16, P17, and P18) exhibited high expression of iCCA$^{phl}$ markers such as S100P and MUC5AC, indicating their origin from large intrahepatic bile ducts. Notably, S100P + cells accounted for 91.14% of total tumor cells from these seven iCCAs and displayed more representative and extensive-expression compared with the other markers (MUC5AC: 42.37%, and MUC6: 22.97%). The 14 iCCAs can be divided into two groups based on S100P expression, which was confirmed by immunohistochemistry (Supplementary Fig 2b, c). For the remaining seven S100P- iCCAs (P09, P10, P12, P13, P14, P15, and P19), they expressed iCCA$^{pps}$ markers NCAM1 and CDH2, which were mutually exclusive with the expression of S100P, confirming the different origins of these tumor cells. NESTIN, which has been proposed as a possible diagnostic biomarker for diagnosing combined hepatocellular carcinoma-intrahepatic cholangiocarcinoma (cHCC-ICC)[15], was mostly expressed in S100P- iCCA cases, suggesting the possible similarities between cHCC-ICC and S100P- iCCA. However, the positive cells of NCAM1 (2.36%) and CDH2 (31.86%) accounted for a very low proportion of the total tumor cells in these seven S100P- iCCAs. To find more representative markers for iCCA$^{pps}$, we searched for genes mutually exclusive with S100P but expressed extensively in iCCA$^{pps}$. Gene such as SPP1 had low expression in S100P + and high expression in S100P- cells, making it potential biomarkers (Fig. 2b and Supplementary Data 2). SPP1, also known as osteopontin (OPN), is highly expressed in a variety of tumors and plays important roles in tumor progression and tumor cell evolution in response to therapy[16,17]. We confirmed that the seven S100P- iCCAs showed high expression of SPP1 both at the cellular (87.17% of S100P- iCCAs' tumor cells) and tissue level (Fig. 2c and Supplementary Fig. 2d). Thus, we divided 14 iCCAs into S100P + SPP1− iCCA$^{phl}$ and S100P-SPP1 + iCCA$^{pps}$ subgroups based on the expression of S100P and SPP1.

According to our scRNA-seq data, most of the tumor cells either expressed S100P (23.95%) or SPP1 (60.05%), while only 10.01% and 5.98% tumor cells showed double negativity or double positivity, respectively (Supplementary Fig. 2e, f). Consistently, we performed the same analyses in Ma et al.'s iCCA scRNA-seq dataset and found that the expression of S100P and SPP1 were mutually exclusive in iCCA cells[17] (Supplementary Fig. 3a, b). We also found a small number of S100P + SPP1 + cells (5.62%) exist in their iCCA cases (Supplementary Fig. 3c, d). Through dimension reduction, we found that the global expression profile of S100P + SPP1 + showed a higher degree of similarity to S100P + SPP1− than the S100P-SPP1 + cells

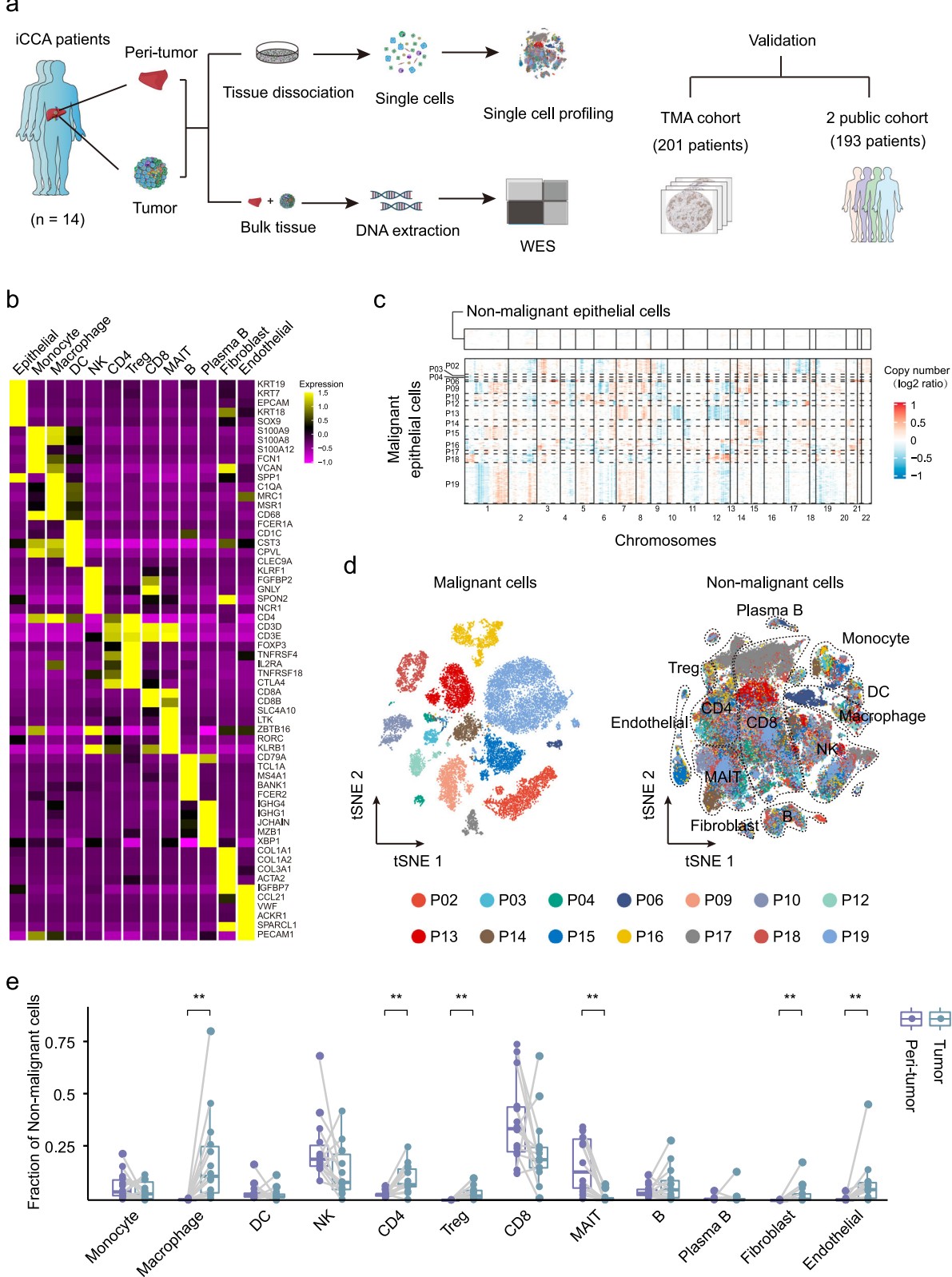

(Supplementary Fig. 3e). Immunohistochemical results revealed that these S100P + SPP1 + cells were mostly present in the invasive regions of cancer nodules in certain iCCA[phl] cases (Supplementary Fig. 3f). Accumulating evidence has revealed SPP1 acts as a significant mediator of modulating tumor invasion and metastasis[18], implying these double-positive cells may be involved in the progression of iCCA.

To further explore whether the expression of S100P and SPP1 in iCCA were mutually exclusive in a larger cohort, immunohistochemistry was performed on a tissue microarray (TMA) containing 201 iCCAs. We found that 92.54% iCCAs can be clearly divided into S100P + SPP1− (33.83%, 68 patients) and S100P-SPP1 + (58.71%, 118 patients) iCCAs, while only 5.97% (12 patients) and 1.49% (three patients) were classified as S100P-SPP1− and

**Fig. 1 ScRNA-seq profiling of 14 iCCAs. a** Schematic representation of the experimental strategy. WES whole-exome sequencing, TMA tissue microarray. Part of the picture was adapted from motifolio.com. **b** Heatmap showing the expression of marker genes in the indicated cell types. **c** Chromosomal landscape of inferred large-scale copy number variations (CNVs) in nonmalignant epithelial cells (top) and malignant cells from 14 iCCA samples. Rows represent individual cells and columns represent chromosomal positions. Amplifications (red) or deletions (blue) were inferred by averaging expression over 100-gene stretches on the respective chromosomes. **d** t-SNE plot of malignant and nonmalignant cells from 14 iCCAs. **e** Boxplot showing the fraction of nonmalignant cells in peri-tumor and tumor. (Peri-tumor $n = 14$, Tumor $n = 14$; **$P < 0.01$; two-sided Wilcoxon matched-pairs signed-rank test; Macrophage: $P = 0.00012$; CD4: $P = 0.0012$; Treg: $P = 0.00012$; MAIT: $P = 0.00012$; Fibroblast: $P = 0.0017$; Endothelial: $P = 0.0012$). The central mark indicates the median, and the bottom and top edges of the box indicate the first and third quartiles, respectively. The top and bottom whiskers extend the boxes to a maximum of 1.5 times the interquartile range. Source data are provided as a Source Data file.

S100P + SPP1 + iCCAs, respectively (Fig. 2d and Supplementary Fig. 4). Our results demonstrated that these S100P + SPP1− iCCAs were more like iCCA[phl] (all were positive for MUC5AC and mucin production), while S100P-SPP1 + iCCAs were more like iCCA[pps] (mostly were negative for MUC5AC and mucin production) by performing the staining of Alcian blue staining (to detect the mucus secreted by mucous tumor cells), immunohistochemical staining of MUC5AC (essential for mucus production), and HE staining (morphology) in these samples (Supplementary Fig. 4). Consistent with the previous study, it's difficult to accurately distinguish iCCA[phl] and iCCA[pps] only from the morphology[19]. Survival analysis revealed that S100P + SPP1− iCCAs had a significantly worse prognosis than S100P-SPP1 + iCCAs ($P = 0.008$, Fig. 2e), which was further confirmed by the multivariate Cox regression analysis (HR, 1.922; 95% CI, 1.257–2.939; $P = 0.003$, Supplementary Data 3). Also, S100P + SPP1− iCCAs significantly correlated with higher CA19-9 ($P < 0.01$), CEA ($P < 0.01$), Ki67 expression ($P = 0.025$), lymph node metastasis ($P = 0.013$), and advanced TNM stage ($P = 0.021$), but negatively correlated with tumor size ($P = 0.019$), HBsAg status ($P < 0.01$), chronic hepatitis ($P = 0.002$) and liver cirrhosis ($P = 0.049$) (Fig. 2f and Supplementary Data 3). The higher percentage of HBsAg positive status, chronic hepatitis, and liver cirrhosis in S100P-SPP1 + iCCAs further support the notion that iCCA[pps] usually develop on a background of chronic liver disease[5].

We further evaluated the effect of S100P and SPP1 in distinguishing iCCA[phl] and iCCA[pps] in two RNA-seq databases of cholangiocarcinoma. We found that 81.48% iCCAs can be divided into two independent groups according to the expression of S100P and SPP1 in Jusakul et al.'s dataset[20] (Fig. 2g and Supplementary Data 4). The S100P-SPP1 + samples almost exclusively exist in iCCA instead of ECC, further supporting their distinct origination (Fig. 2h). Survival analysis showed that the prognosis of S100P + SPP1− iCCAs were significantly worse than S100P-SPP1 + iCCAs ($P < 0.01$, Fig. 2i). Similar results were obtained from Job et al.'s dataset[21] (Supplementary Fig. 5a, b).

Analysis of the WES data found that S100P + SPP1− iCCAs tended to have more *TP53* (4/14), *SYNE1* (3/14), and *EPHA2* (3/14) mutations, while S100P-SPP1 + iCCAs harbored more *BAP1* (3/14) mutations, which was consistent with previous studies[10,20] (Supplementary Fig. 5c, d). We also found that the DNA methylation level of *S100P* in S100P + SPP1− was significantly lower than that in S100P-SPP1+, while no apparent difference was observed in CNVs[20], indicating potential epigenetic regulation of *S100P* in these two iCCA subtypes (Supplementary Fig. 5e, f). Taken together, these results indicate that S100P and SPP1 are two optimal biomarkers for distinguishing iCCA[phl] and iCCA[pps], which can effectively divide the iCCA patients into two subtypes with different cell origins and clinicopathological characteristics.

**Molecular profiles and transcription networks of S100P + SPP1− and S100P-SPP1 + iCCAs.** The presence of two main subgroups of malignant cells in iCCA prompted us to investigate their unique gene expression profiles. We first evaluated their

intratumor heterogeneity (ITH) at the genomic and single-cell transcriptome levels. The results showed no significant difference in genomic ITH, but a significantly higher transcriptomic ITH in S100P + SPP1− iCCAs (Fig. 3a). This was consistent with a previous study that higher transcriptomic ITH predicted poor survival[11]. Subsequently, we identified 755 differentially expressed genes between these two groups of malignant cells ($|logFC| > 1.5$ and $P < .01$, Supplementary Fig. 6a and Supplementary Data 5). Genes upregulated in S100P-SPP1 + cells were mainly enriched in the regulation of coagulation and complement activation, which were involved in hepatocyte function (Fig. 3b). These cells presented high expression of hepatocyte-specific genes such as *SERPINE2*, *APOB*, and *CPB2*, further supporting their hepatocyte-like differentiation (Supplementary Fig. 6b). In contrast, genes upregulated in S100P + SPP1− cells were related to mucus secretion, protein localization to the endoplasmic reticulum (ER), and epithelial structure maintenance. Remarkably, we found that *PSCA*, which encodes a tumor antigen and is upregulated in prostate[22] and bladder[23] cancers, was highly expressed in S100P + SPP1− iCCAs, making it a promising candidate for immunotherapy of iCCA[phl] (Supplementary Fig. 6c).

We further applied SCENIC analysis to characterize transcription networks between S100P + SPP1− and S100P-SPP1 + cells[24]. The results showed that transcription factors such as *ATF3*, *CREB5*, *MEIS2*, and *EGR1* were upregulated in S100P-SPP1 + cells, while S100P + SPP1− cells showed upregulation of transcription factors like *CREB3L1*, *PPARG*, *CDX2*, and *HOXB7* (Fig. 3c and Supplementary Fig. 6d). Survival analysis from Jusakul et al.'s dataset[20] showed that transcription factors that highly expressed in iCCA[phl] (*PPARG*, *MECOM*, *HOXB7*, *IRF7*, *FOXA3*) and iCCA[pps] (*ONECUT1*, *HNF1B*, *MEIS2*), were associated with worse and better prognosis, respectively (Supplementary Fig. 6e). Notably, the SCENIC analysis revealed that *CREB3L1*, which is induced by ER stress and contributes to maximal induction of the unfolded protein response[25], was a potential transcription factor regulating *S100P*. Also, *CREB3L1* expression strongly and positively correlated with *S100P* expression ($r = 0.58$, $p < 2.2e-16$, Fig. 3d). To determine whether *S100P* is a direct target of *CREB3L1*, we performed a dual-luciferase report assay and found that the *S100P* promoter activity was markedly increased in a dose-dependent manner after overexpression of *CREB3L1* (Fig. 3e). Transwell assays showed that *CREB3L1* knockdown significantly weakened the invasion capacity of HuCCT1 and RBE cells (Fig. 3f, g). RNA-seq analysis showed that *CREB3L1* not only modulated the expression of *S100P*, but also affected the expression of various upregulated genes in S100P + SPP1− cells, such as *OASL*, *RCN3*, and *OAS1* (Fig. 3h). Pathway analysis indicated that *CREB3L1* was involved in co-translational protein targeting to membrane, the establishment of protein localization to ER, and actin filament reorganization (Fig. 3i). Together, these results reveal the distinct transcriptional profiles of S100P + SPP1− and S100P-SPP1 + cells, identifying CREB3L1 as a potential transcriptor of S100P that promotes invasion of iCCA[phl].

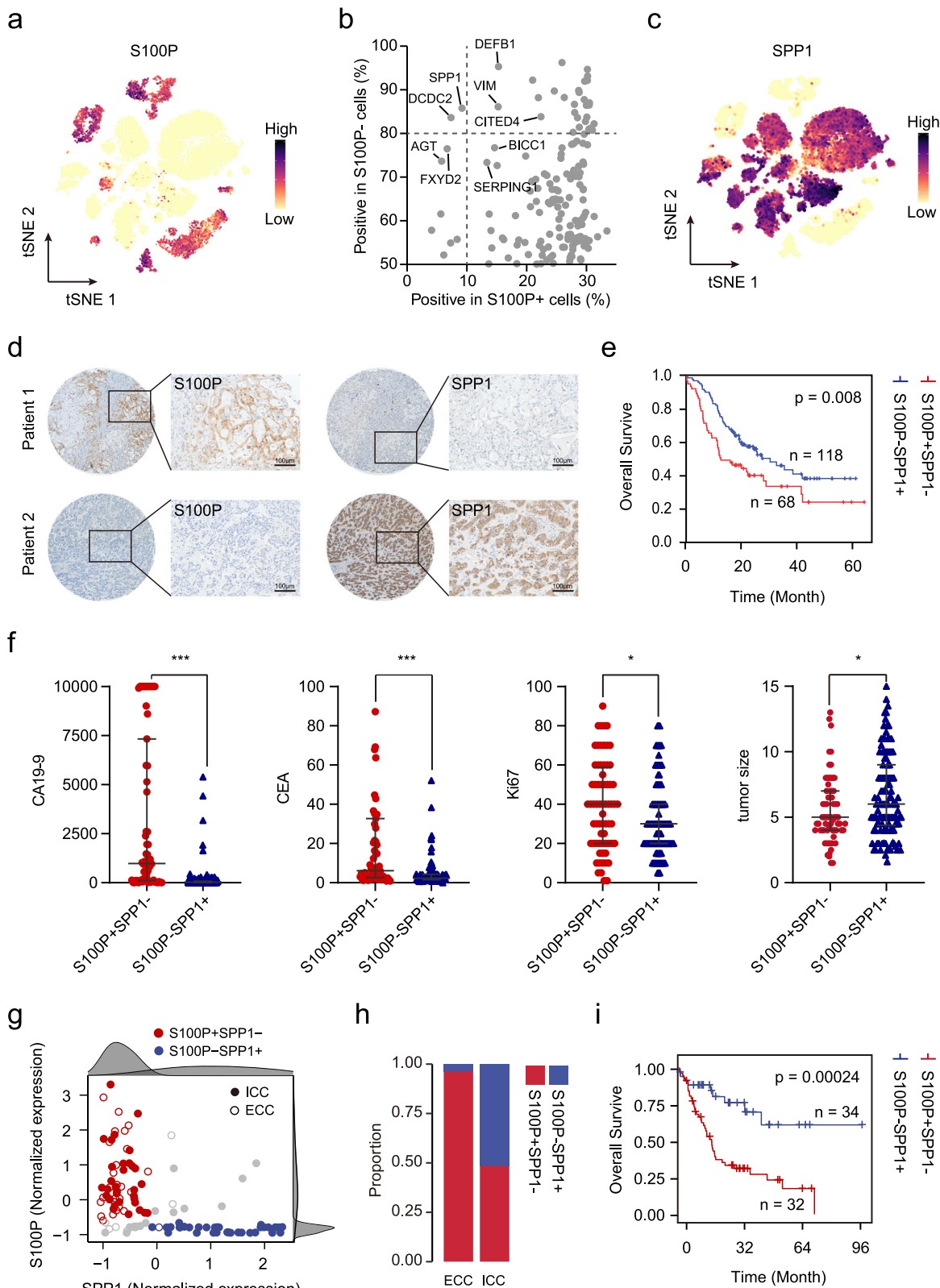

**Different polarization of infiltrated macrophages in iCCA[phl] and iCCA[pps]**. Despite studies have profiled the tumor immune microenvironment of iCCA by scRNA-seq[11,12], the difference of immune landscape between iCCA[phl] and iCCA[pps] remains unclear. First, we evaluated the infiltration of T cells, B cells, NK cells, and macrophages in 186 iCCAs from the TMA cohort by immunostaining. Results showed that more CD3[+] T cells

($P < 0.01$) and CD56[+] NK cells ($P < 0.01$) were infiltrated in S100P-SPP1 + iCCA[pps] (118 patients) compared to S100P + SPP1− iCCA[phl] (68 patients). Further analysis of T cell subsets revealed that iCCA[phl] harbored increased CD8[+] T cells while decreased CD4[+] T cells than iCCA[pps]. In addition, iCCA[phl] displayed significantly higher PD1[+]CD8[+] T cells infiltration than iCCA[pps] ($P < 0.01$), while no significant difference in

**Fig. 2 iCCA can be classified into two subtypes according to the expression of S100P and SPP1. a** t-SNE plot showing the expression level of S100P in malignant cells. **b** Proportion of positive cells with gene expression in S100P + (x-axis) and S100P- cells (y-axis). **c** t-SNE plot showing the expression level of SPP1 in malignant cells. **d** Representative images of immunohistochemical expression of S100P and SPP1 in iCCAs from TMA cohort (*n* = 201). Patient 1: S100P + SPP1−, Patient 2: S100P-SPP1+. Scale bar, 100 μm. The experiment was repeated once with similar results. **e** Kaplan–Meier plot of the S100P + SPP1− and S100P-SPP1+ based on TMA data. Two-sided log-rank test. **f** The scatter diagrams showing the differences in carbohydrate antigen 19-9 (CA19-9, S100P + SPP1− *n* = 63, S100P-SPP1+ *n* = 114), carcinoembryonic antigen, (CEA, S100P + SPP1− *n* = 63, S100P − SPP1+ *n* = 115), Ki67 (S100P + SPP1− *n* = 68, S100P−SPP1+ *n* = 118), and tumor size (S100P + SPP1− *n* = 68, S100P-SPP1+ *n* = 118) between the two groups (*$P < 0.05$; ***$P < 0.001$; two-sided Mann–Whitney *U*-test; CA19-9: $P < 0.0001$; CEA: $P < 0.0001$; Ki67: $P = 0.025$; tumor size: $P = 0.019$). Data were presented as median with interquartile range. **g** Scatterplot of S100P and SPP1 expression in Jusakul et al. dataset[20]. A Gaussian mixture model with two mixture components was used to identify S100P +/− and SPP1 +/− patients (right and top distribution curves). Solid circles represent iCCA and open circles represent extrahepatic cholangiocarcinoma (ECC). Red represents S100P + SPP1- while blue represents S100P-SPP1+. **h** Graphical representation of the proportion of S100P + SPP1- and S100P-SPP1+ in iCCA and ECC. **i** Kaplan–Meier plot of the S100P + SPP1− and S100P-SPP1+ based on Jusakul et al. dataset[20]. Two-sided log-rank test. Source data are provided as a Source Data file.

FOXP3+CD4+ Treg cells infiltration (Supplementary Fig. 7a). Although there was no significant difference in CD68+ macrophages and CD20+ B cells, more CD68+CD206+ macrophages were found to be infiltrated in S100P + SPP1− iCCA^phl ($P < 0.01$) (Supplementary Fig. 7b, c). Then, we focused on macrophages to evaluate distinct macrophage subsets infiltrated in the two subtypes of iCCAs.

A total of six clusters present in the myeloid lineage with the expression of specific marker genes, including one monocyte (Mono_FCN1), two macrophages (Macro_c1_SPP1 and Macro_c2_CCL18), and three DCs (DC_c1_CD1C, DC_c2_XCR1, and DC_c3_CD1A) (Fig. 4a, b and Supplementary Data 6). Macrophages and CD1a + DCs (DC_c3_CD1A) were significantly enriched in tumors compared with paired non-tumor tissues, while monocytes, CD1c+ DCs (DC_c1_CD1C), and cDC1 DCs (DC_c2_XCR1) showed the opposite trend (Supplementary Fig. 7d). Indeed, we observed that SPP1+ macrophages, which have been reported in colon cancer and closely interact with cancer-associated fibroblasts (CAFs)[26], were more infiltrated in S100P-SPP1+ iCCA^pps, while CCL18+ macrophages, which were abundant in advanced hepatocellular carcinoma[27], were mostly infiltrated in S100P + SPP1− iCCA^phl (Fig. 4c, d). Though both macrophages subsets have been defined as tumor-associated macrophages, they varied in signaling pathways and metabolic features[28] (Supplementary Fig. 8a, b). Consistently, we found that SPP1+ macrophages showed an increased level of oxidative phosphorylation and glycine, serine, threonine, and tyrosine metabolism, while CCL18+ macrophages had elevated cytokine–cytokine receptor interaction, nitrogen, and riboflavin metabolism (Supplementary Fig. 8c). By calculating pro-/anti-inflammatory and M1/M2 polarization scores[29], we found that SPP1+ macrophages were more potent in both pro- and anti-inflammatory responses and skewed toward M1 polarization (Fig. 4e, f). In contrast, CCL18+ macrophages showed a dominant M2-like phenotype with the high expression of *CD163*, *MARCO*, and *CSF1R*, suggesting their stronger tumor-promoting role than SPP1+ macrophages (Supplementary Fig. 8d). Immunostaining on the TMA cohort further confirmed that SPP1+CCL18− macrophages were more abundant in S100P-SPP1+ iCCA^pps, while SPP1−CCL18+ macrophages were mostly enriched in S100P + SPP1− iCCA^phl (Fig. 4g, h), which were again validated by the results from Jusakul et al.'s dataset[20] (Supplementary Fig. 8e). Together, these results indicate that iCCA^phl has a unique immune ecosystem, with increased CCL18+ macrophages, reduced CD3+ T and CD56+ NK cells as compared with iCCA^pps.

**iCCA^pps contains tumor cells at different status of differentiation.** The expression of ALB is generally considered a marker of hepatocytes. Several studies have demonstrated the expression of ALB in iCCA, but the features of these ALB + tumor cells are still unclear[8,30,31]. Here, we detected a group of ALB-expressing tumor cells at the single-cell level, most of which (79.4%) were present in the S100P-SPP1 + iCCA^pps (Supplementary Fig. 9a–c). Due to the different origins of iCCA^phl and iCCA^pps, we here only focused on these seven S100P-SPP1 + iCCA^pps to explore their heterogeneity. By comparing the gene expression profiles of ALB + and ALB- cells, we found that ALB- cells highly expressed *ID3*, which negatively regulates the basic helix-loop-helix and is involved in cell differentiation, and neoplastic transformation[32] (Fig. 5a, Supplementary Fig. 9d, and Supplementary Data 7). ALB + cells highly expressed hepatocyte-specific genes such as *CPB2*, *ASGR1*, *FGA*, as well as cholangiocyte markers *KRT19*, *KRT18*, and *EPCAM*, but did not express *AFP*, a marker of hepatic progenitor cells (Fig. 5b and Supplementary Fig. 9e). Genes that are highly expressed in ALB + cells were mainly involved in hepatocyte-specific processes, such as complement activation, detoxification, fatty acid catabolic process, and bile acid secretion, suggesting their hepatocyte differentiation (Supplementary Fig. 9f). SCENIC analysis showed that genes specifically upregulated in ALB + cells were regulated by *NR5A2*, *BATF*, and *NFIA* (Supplementary Fig. 9g). In contrast, ID3 + cells highly expressed genes such as *MDK*, *ZEB1*, and *LGR5* that play important roles in tumor stemness[33–36]. The SCENIC analysis predicted that transcription factors *SOX11*, *PAX2*, *IRX2*, *IRX3*, *FOXC1*, and *EN2* were responsible for genes upregulated in these cells.

Previous studies have designated ID3 + cells as hepatoblasts which could give rise to both hepatocytes and cholangiocytes[37]. To reveal the differentiation process in iCCA, we explored the gene expression patterns along this transition by trajectory analysis. Tumor cells from P09 and P10 were selected for this analysis as they contained a comparable number of ALB + and ID3 + cells (Fig. 5c and Supplementary Fig. 9h). We found that ALB + cells were mainly located at the terminal of this trajectory and genes involved in the regulation of coagulation, ER lumen, and response to ER stress were increased gradually along the trajectory (Fig. 5d). Also, the expression of *MKI67* showed the same trend as *ALB*, implying an increased proliferation capacity of ALB + cells. ID3 + cells located opposite to ALB + cells in the trajectory and were enriched for pathways in the collagen-containing extracellular matrix and negative regulation of cell adhesion. For example, the expression of *COL12A1*, which encodes the alpha chain of type XII collagen and is overexpressed in several cancer types[38,39], decreased gradually along the transition from ALB- cells to ALB + cells (Fig. 5e).

By evaluating the expression of 16 identified marker genes of ID3 + and ALB + cells in Jusakul et al.'s dataset[20], we validated that S100P-SPP1 + patients can also be clearly divided into two subclasses with a mutually exclusive expression of 16 genes

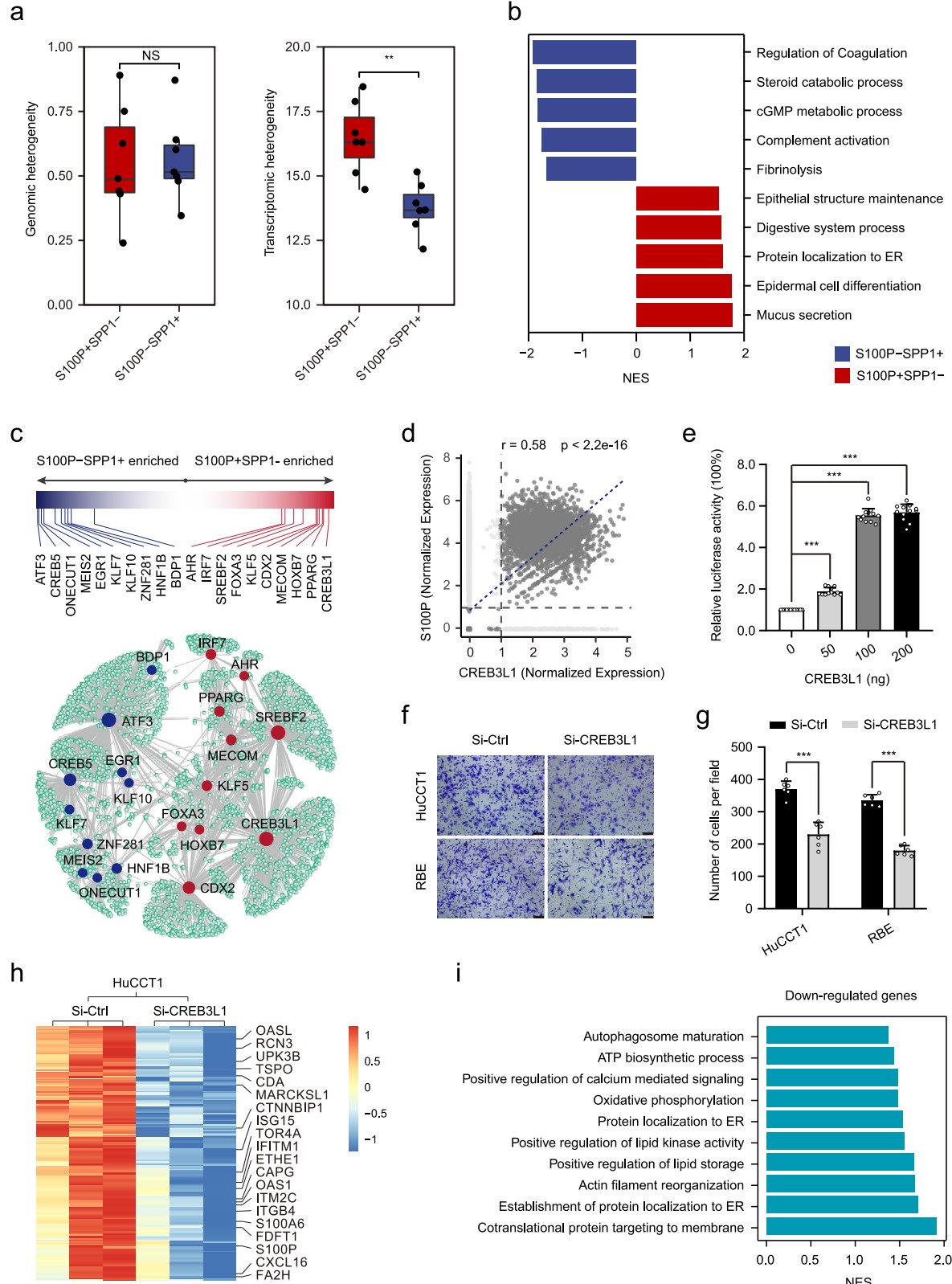

(Fig. 5f). In addition to the exclusivity between *ALB* and *ID3*, a significantly negative correlation between *ID3* and *MKI67* expression was also observed, suggesting the slow proliferation of these tumors (Fig. 5g). Taken together, these results demonstrate that iCCA^PPS is a heterogeneous tumor with tumor cells at the various status of differentiation such as hepatocyte differentiation or stemness.

**ID3 + tumor cells indicate abundant stroma components and worse prognosis in iCCA^PPS.** We next explored the clinical and histological characteristics of ID3 + iCCA^PPS. By immunostaining, we find that ID3 was predominantly expressed in the nucleus of tumor cells located in the tumor center and were surrounded by rich stromal components (Fig. 6a). To further explore the relationship between *ID3* expression and tumor stroma, we

**Fig. 3 Different gene expression profiles between S100P + SPP1− and S100P-SPP1+ cells. a** Boxplot of the genomic heterogeneity (left) and transcriptomic heterogeneity (right) of S100P + SPP1−($n = 7$) and S100P-SPP1 + ($n = 7$) iCCAs. (**$P < 0.01$; two-sided Wilcoxon rank-sum test; Transcriptomic heterogeneity: $P = 0.0041$; NS not significant). The central mark indicates the median, and the bottom and top edges of the box indicate the first and third quartiles, respectively. The top and bottom whiskers extend the boxes to a maximum of 1.5 times the interquartile range. **b** Top enriched pathways for genes with specific expression in S100P + SPP1− and S100P-SPP1 + cells. **c** Network representation of selected differentially expressed transcription factors between S100P + SPP1− and S100P-SPP1 + cells, as analyzed by SCENIC. Transcription factors in S100P + SPP1− are shown in red; transcription factors in S100P-SPP1 + are shown in blue. Bar graph showing the difference score for the selected set of differentially expressed transcription factors in S100P + SPP1− (red) and S100P-SPP1 + (blue). **d** Scatterplot showing the correlation of CREB3L1 expression (x-axis) with S100P expression (y-axis). Correlation is evaluated by the Spearman correlation coefficient. **e** The relative luciferase activity in HEK-293T cells following co-transfection with plasmid containing S100P promoter and increasing doses of the CREB3L1 expression vector (***$P < 0.001$; two-sided student's $t$-test; CREB3L1 50 ng: $P < 0.0001$; 100 ng: $P < 0.0001$; 200 ng: $P < 0.0001$; $n = 12$ biologically independent samples). **f, g** Representative images of the Transwell invasion assay (**f**) and a statistical histogram (**g**) (***$P < 0.001$; two-sided student's $t$-test; HuCCT1: $P < 0.0001$; RBE: $P < 0.0001$; $n = 6$ biologically independent samples). Scale bar, 100 μm. **h** Heatmap displaying expression levels of differentially expressed genes in Si-CREB3L1 versus Si-Ctl in HuCCT1 cells. **i** Top enriched pathways for downregulated genes in Si-CREB3L1 HuCCT1 cells. Si-Ctl Small interfering control (**f, g**, and **h**); NES normalized enrichment score (**i**). Error bars of (**e** and **g**) represent the means ± SD. Source data are provided as a Source Data file.

analyzed the correlation between *ID3* expression and CAFs in two public databases[20,21]. Results showed that *ID3* expression positively correlated with CAFs' gene signature, such as *PDGFRB*, *COL1A1*, and *PDPN* (Fig. 6b and Supplementary Fig. 9i).

Since CAFs play important roles in tumor progression and chemoresistance[40], we speculated that ID3 expression was related to iCCA prognosis. We selected 118 S100P-SPP1 + iCCA^PPs from our TMA cohort to explore the prognostic values of ID3 + tumor cells and PDGFRβ + stromal cells (most of which were CAFs) (Fig. 6c). As expected, the proportion of CK19 + ID3 + tumor cells positively correlated with the proportion of CK19-PDGFRβ + cells ($r = 0.46$, $P < 0.001$), while the proportion of CK19 + ID3- tumor cells negatively correlated with the proportion of CK19-PDGFRβ + cells ($r = −0.46$, $P < 0.001$) (Fig. 6d). Survival curves indicated that the proportion of CK19 + ID3 + tumor cells ($P = 0.016$) and CK19-PDGFRβ + cells ($P = 0.005$) both significantly correlated with poor prognosis in iCCA^PPs (Fig. 6e). Thus, these results demonstrate that ID3 + cells commonly correlated with the presence of CAFs and patient survival in iCCA^PPs.

## Discussion

iCCAs can be divided into two main histological subtypes, iCCA^phl and iCCA^PPs, according to the tumor anatomical location and the origin of tumor cells. In this study, we generated scRNA-seq profiles of 14 primary iCCAs and identified SPP1 as a representative marker for iCCA^PPs. We found that 92.5% iCCAs can be classified as iCCA^phl and iCCA^PPs according to the expression of S100P and SPP1, and there are significant differences in clinicopathological characteristics, gene regulatory networks, and immune infiltration between these two iCCA subtypes. Moreover, we confirmed the presence of tumor cells at various differentiation in iCCA^PPs at the single-cell level (Fig. 7).

Cholangiocarcinoma can be divided into iCCA (which arises above the second-order bile ducts) or ECC (including perihilar CCA and distal CCA) according to the tumor location in the biliary tree. Compared to iCCA^PPs, iCCA^phl comprises mucin-producing columnar tumor cells and has high invasiveness and high expression of S100P, which is more similar to ECC[8]. We here identified S100P + SPP1− cells, which were mostly present in iCCA^phl, highly expressed mucus-related genes such as *MUC5AC*, and *MUC6* at the single-cell level. Mucins synthesis begins in the ER and they are extremely susceptible to misfolding due to their large sizes and structure complexity, which can eventually lead to ER stress[41]. We indeed observed many genes associated with mucins synthesis or ER stress upregulated in iCCA^phl, such as *XBP1*[42], *AGR2*[43], and *CREB3L1*[25], which may be

involved in the progression of this subtype of iCCA. We also found that despite S100P + SPP1− iCCA^phl often had smaller tumor size, it had more lymph node metastases, and higher levels of CA19-9, Ki67, and CEA compared with S100P-SPP1 + iCCA^PPs. This further suggested that there are marked differences in clinical characteristics between these two iCCA subtypes. Notably, there were also significant differences in the infiltration of several important immune cells between them. S100P + SPP1− iCCA^phl had less CD3+ T and CD56+ NK cells, but more CCL18+ macrophage infiltration than S100P-SPP1 + iCCA^PPs, indicating its dampened anti-tumor immune response that may contribute to the higher invasive potential.

SPP1 is considered to play a cancer-promoting role and is often associated with a worse prognosis in various tumors, but its prognostic significance in iCCA is still controversial[44,45]. One important reason for this inconsistency is that the classification of iCCA is not properly considered. iCCA^PPs is believed to originate from mucin-negative cuboidal cholangiocytes or ductules containing hepatic progenitor cells. It has been reported that *CDH2* and *NCAM1*, are representative markers of these iCCAs[3,5]. Based on our results, the expression of *SPP1* is mutually exclusive with *S100P*, showing a better specificity and sensitivity than *CDH2* or *NCAM1* as a marker of iCCA^PPs. S100P-SPP1 + iCCA^PPs had less lymph node metastasis, larger tumor volume, and better prognosis than S100P + SPP1− iCCA^phl. It has been noted that the occurrence of these two subtypes of iCCA was related to different pathogenic factors[46–48]. The iCCA^PPs usually develop on a background of chronic viral hepatitis or liver cirrhosis compared with iCCA^phl, which often develop under primary sclerosing cholangitis (PSC) or liver fluke infection status[3]. Our results revealed that there were significantly higher percentages of HBsAg positive status, chronic hepatitis, and liver cirrhosis in S100P-SPP1 + iCCA^PPs than in S100P + SPP1− iCCA^phl, further highlighting their distinct pathogenic background. One research has reported that iCCA with cholangiolocellular differentiation highly expressed *CRP* and *CDH2*, while iCCA without cholangiolocellular differentiation highly expressed *TFF1* and *S100P*. The two groups of iCCAs showed significant differences in clinicopathological characteristics and patient outcomes[9]. The results from this study are very similar to the findings of our study. S100P-SPP1 + iCCA^PPs showed high expression of *CRP* and *CDH2*, which correspond to the iCCAs with cholangiolocellular differentiation. Studies have revealed that iCCA^PPs often occur in the background of chronic hepatitis or liver cirrhosis[5]. We observed SPP1+ macrophages, which has been reported involving in liver inflammation and fibrosis[49], were highly infiltrated in iCCA^PPs, indicating that these macrophages may be involved in the occurrence and development of iCCA^PPs. It

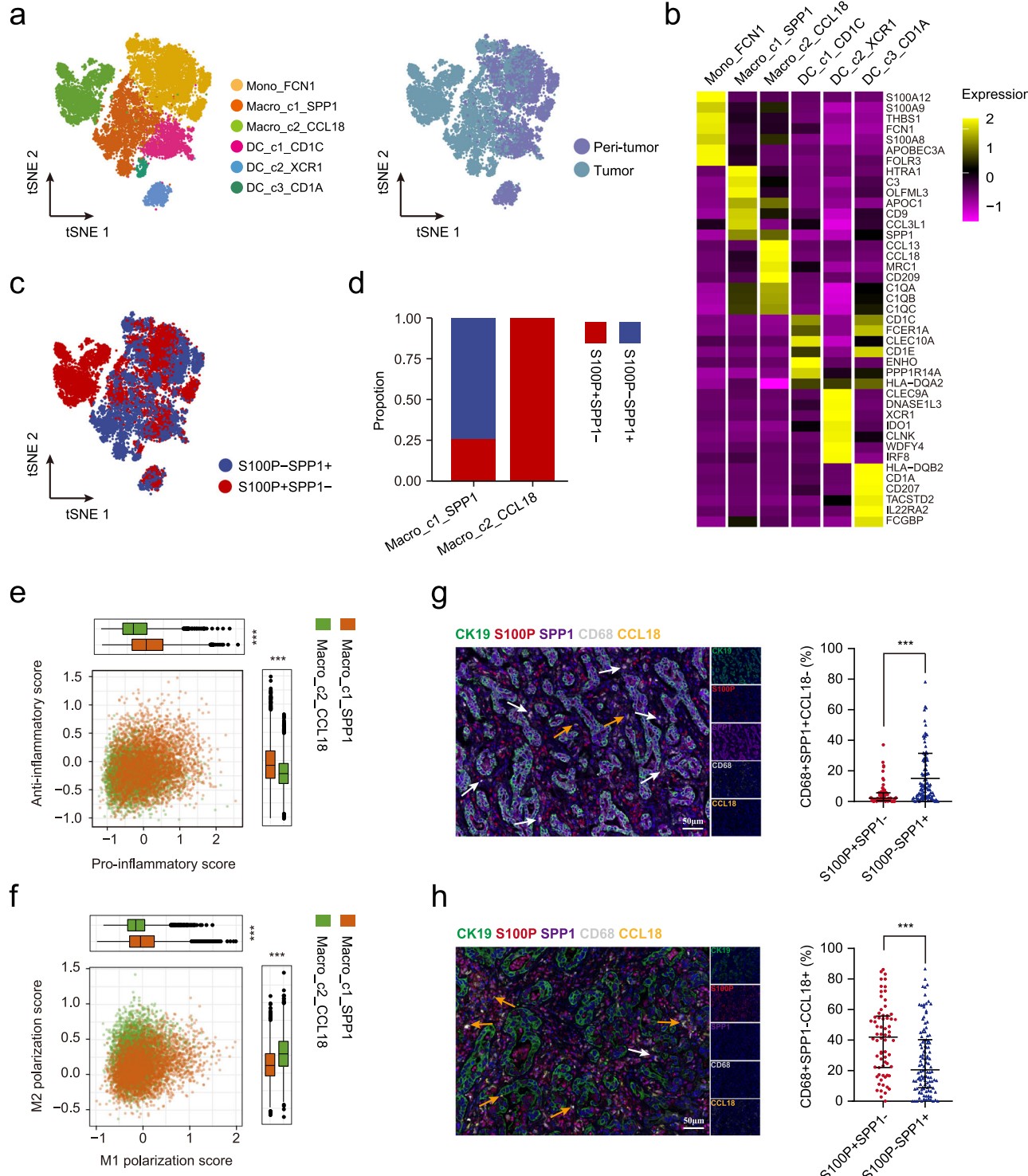

**Fig. 4 Two different subsets of macrophages infiltrated in iCCA[phl] and iCCA[pps]. a** The t-SNE plot showing the subtypes of myeloid cells derived from iCCA peri-tumor and tumor. **b** Heatmap showing the expression of marker genes in each subtype of myeloid cells. **c** t-SNE plot of myeloid cells from S100P + SPP1− (red dots) and S100P-SPP1 + (blue dots). **d** Bar plot showing the proportion of macrophage subsets from S100P + SPP1− and S100P-SPP1+. **e**, **f** Scatterplots showing pro-/anti-inflammatory scores (**e**) and M1/M2 scores (**f**) for two macrophage subsets. Macro_c1_SPP1, $n = 4016$ cells; Macro_c2_CCL18, $n = 3447$ cells. (***$P < 0.001$; two-sided Wilcoxon rank-sum test; Anti-inflammatory score: $P < 2.22e-16$; Pro-inflammatory score: $P < 2.22e-16$; M2 polarization score: $P < 2.22e-16$; M1 polarization score: $P < 2.22e-16$). The central mark indicates the median, and the bottom and top edges of the box indicate the first and third quartiles, respectively. The top and bottom whiskers extend the boxes to a maximum of 1.5 times the interquartile range. **g**, **h** Representative mIHC images (left) and statistical graphs (right) to show the distribution of CD68[+]SPP1[+]CCL18[−] and CD68[+]SPP1[-] CCL18[+] macrophages in S100P[+]SPP1− (**g**) and S100P−SPP1 + (**h**), respectively: CK19 (green), S100P (red), SPP1 (purple), CD68 (white), CCL18 (yellow), and DAPI (blue) (S100P + SPP1− $n = 68$, S100P-SPP1 + $n = 112$). White arrows (CD68 + SPP1 + CCL18−), yellow arrows (CD68 + SPP1−CCL18+). (***$P < 0.001$; two-sided Mann–Whitney $U$-test; CD68 + SPP1 + CCL18− (%): $P < 0.0001$; CD68 + SPP1-CCL18 + (%): $P < 0.0001$). Data were presented as median with interquartile range (**g** and **h**). Scale bar, 50 μm. Source data are provided as a Source Data file.

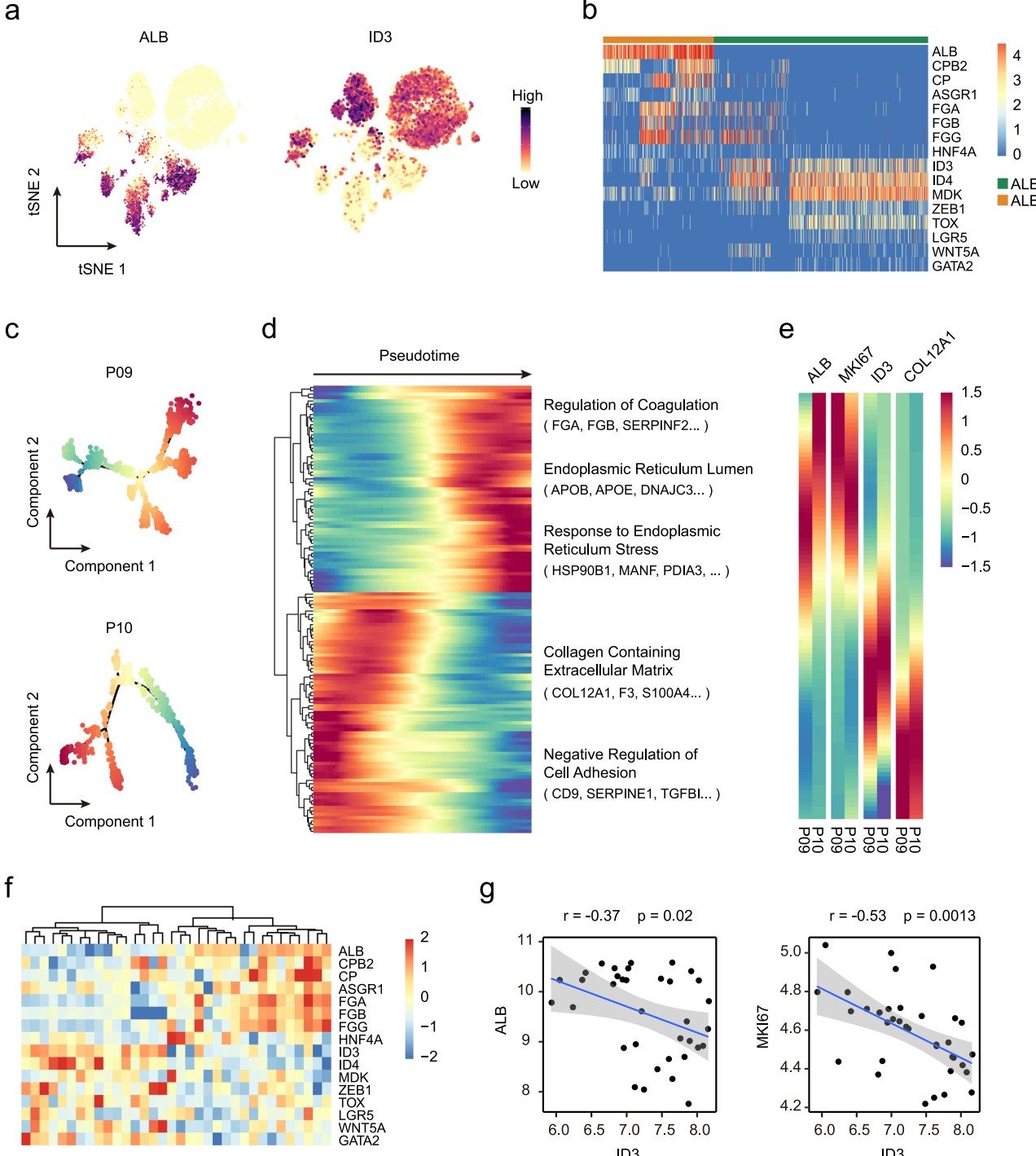

**Fig. 5 Tumor cells at different status of differentiation exist in S100P-SPP1 + iCCAs. a** t-SNE plot showing expression levels of ALB and ID3 in 7 S100P-SPP1 + iCCAs. **b** Heatmap showing expression levels of differentially expressed genes (rows) between ALB + and ALB- S100P-SPP1 + tumor cells (columns). **c** Trajectory of tumor cells from P09 and P10 separately in a two-dimensional state-space defined by Monocle. **d** Differentially expressed genes along the pseudo-time were clustered hierarchically into two profiles. The representative gene functions and pathways were shown. **e** Heatmap showing expression of representative genes. Color key from blue to red indicates relative expression levels from low to high. **f** Heatmap of ALB + and ALB- specific genes (rows) and hierarchical clustering result in 34 S100P-SPP1 + iCCA (columns) from Jusakul et al. dataset[20]. **g** Correlation between expression of ID3 and expression of ALB and MKI67. Blue line represents the linear regression curve. The gray band represents the 95% confidence interval of the regression line. Correlation is evaluated by the two-sided Spearman correlation coefficient. Source data are provided as a Source Data file.

should be noted that two isoforms of SPP1 (iOPN and sOPN) with distinct functions could be generated by an alternative translation that we could not determine whether the form of SPP1 expressed by SPP1 + macrophages was the same as that of SPP1 + tumor cells, which needs to be further explored[16].

Heterogeneity in tumor cell differentiation was observed in iCCA because of the complicated cell origin and formation. In the present study, two major subsets of tumor cells, ALB + and ID3 + tumor cells were identified in iCCA[pps]. The expression of ALB mRNA has been detected by in situ hybridization in about

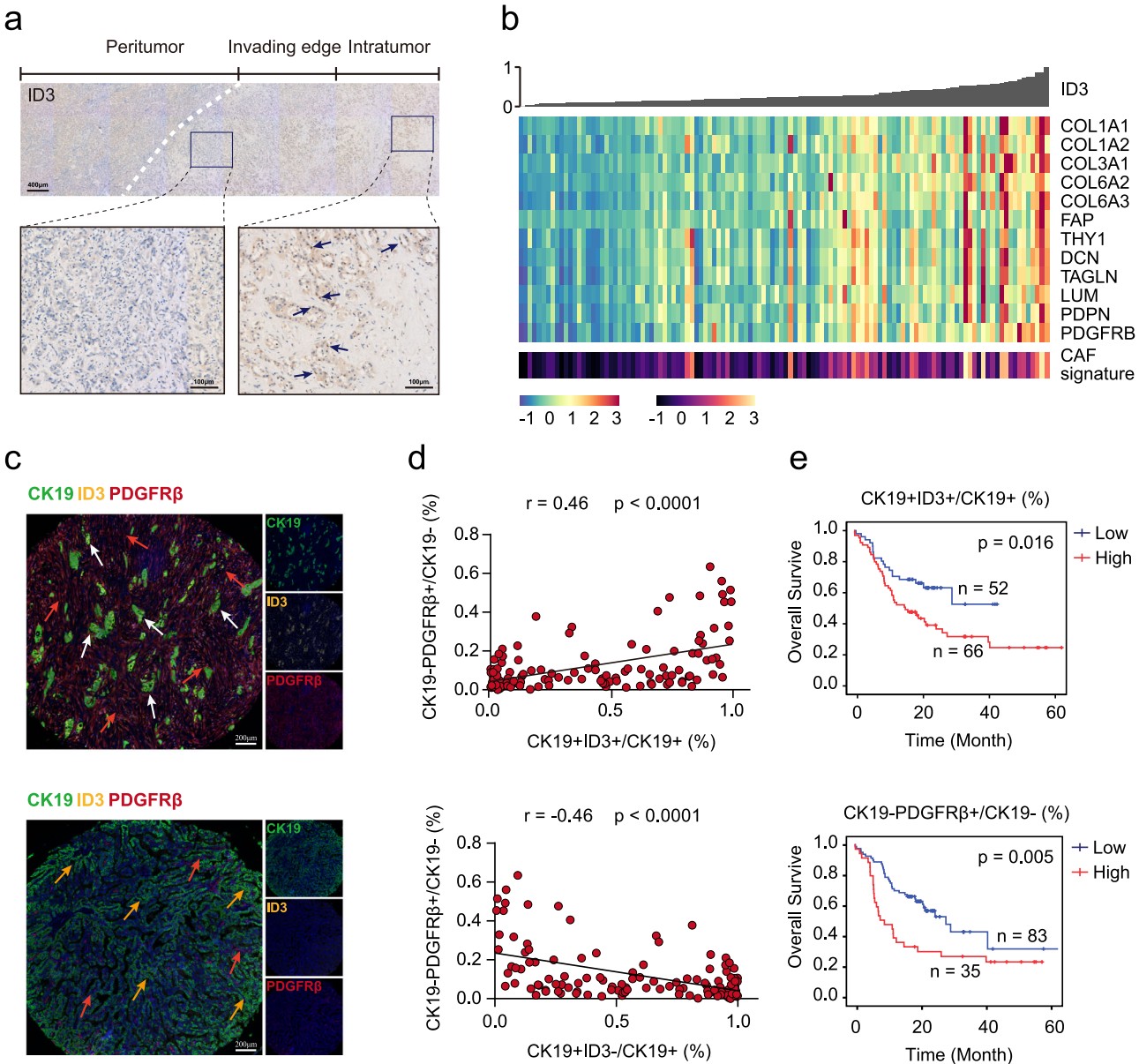

**Fig. 6 Prognostic significance of CK19 + ID3 + tumor cells in S100P-SPP1 + iCCAs. a** Representative immunostaining of ID3 in the indicated S100P-SPP1 + iCCAs. ID3 + tumor cells were predominantly located in the intratumor region. Scale bar, 400 μm (up) and 100 μm (down). Images were collected from 17 additional iCCA slides that contained both tumor and corresponding paracancerous tissues. The experiment was repeated once with similar results. **b** Correlation between ID3 expression and CAFs. iCCA from Jusakul et al.'s dataset[20] were ordered by their ID3 expression level as shown by bar plot (top). Heatmap (middle) showing expression levels of selected CAF markers (rows) for each tumor (columns). Colored bar (bottom) showing the CAFs score estimated by MCP-Counter of each tumor. **c** Representative mIHC images showing the distribution of CK19 + ID3 + , CK19 + ID3- tumor cells and CK19-PDGFRβ + cells in S100P-SPP1 + iCCA (n = 118) from TMA cohort: CK19 (green), ID3 (yellow), PDGFRβ (red), and DAPI (blue). White arrows (CK19 + ID3 + ), yellow arrows (CK19 + ID3−), red arrows (CK19-PDGFRβ+). The experiment was repeated once with similar results. Scale bar, 200 μm. **d** Correlation analysis between the proportion of CK19 + ID3 + (up) and CK19 + ID3- (down) within CK19 + tumor cells and the proportion of CK19-PDGFRβ + cells within CK19− cells per core, respectively. (Two-sided spearman correlation coefficient). **e** Kaplan–Meier analysis of overall survival (OS) in S100P-SPP1 + iCCA tumors according to the proportion of CK19 + ID3 + within CK19+ tumor cells (up) and CK19-PDGFRβ + within CK19− cells (down) in the TMA cohort. Two-sided log-rank test. Source data are provided as a Source Data file.

40% of all iCCAs[50], but the specific biology of these ALB+ cells is still not clear. The results of our study showed that these ALB+ cells have the characteristics of hepatocyte differentiation. However, these cells also expressed *EPCAM* and *KRT19*, indicating that these may be hepatocyte-like cells in the early stage of differentiation rather than mature hepatocytes. The stem-like ID3+ cells coexisting in iCCA[PPs] may be the precursor cells of ALB + cells. There are several reasons for this conjecture. First, these ID3 + cells highly expressed many stemness-related genes, such

as *ID4*, *MDK*, *ZEB1*, and *LGR5*. Of note, it has been reported that the expression of *ID3* and *LGR5* could promote stem cell features in iCCA[33,36]. Second, a previous study has identified ID3 + cells at the early stages of development in human and mouse fetal livers, which are able to differentiate into both hepatocytes or cholangiocytes[37]. Therefore, the presence of ID3 + cells may be one of the reasons for the diversity of iCCA[PPs]. Additionally, we found that ID3 + cells were generally located in the interior area of the iCCA[PPs] and positively correlated with the CAF content.

**Fig. 7 Schematics for the classification of iCCA. Two major subtypes of iCCA were identified in this study.** Morphological features, cellular component, immune infiltration, and prognosis varied significantly between these two iCCA subtypes. Part of the picture was adapted from motifolio.com.

The location of these ID3 + cells and the presence of a large amount of CAFs surrounding them may be an important reason for the poor prognosis of this type of iCCA.

A few limitations of the current study should not be ignored. There were 1.49% S100P + SPP1+ and 5.97% S100P-SPP1− iCCAs in our validation cohort. We did not analyze the clinicopathological features of these iCCAs because of their small number. Also, due to the small number of S100P + SPP1 + and S100P-SPP1− cells in scRNA-seq data, we could not evaluate the molecular characteristics of these two types of tumor cells at the single-cell level accurately. Therefore, future studies with a larger sample size containing these two iCCA subtypes may help to resolve this issue. Furthermore, the lack of functional data in our study restricts our understanding regarding the molecular mechanisms underlying the tumorigenesis of these two iCCA subtypes. Further animal experiments may shed light on this issue and validate our results in the future.

In summary, our findings suggest that iCCA^phl and iCCA^pps have distinct cell origins. Nevertheless, it is often difficult or impossible to accurately distinguish them by conventional methods, such as evaluating their cellular morphology, architectural features or mucin productivity, because a certain proportion of iCCAs contain mixtures of the large duct and small duct types and also displayed atypical histology and the combined detection of the expression of multiple tissue markers may facilitate their distinguishment. We here suggest two markers, S100P, and SPP1 differentiate between iCCA^phl and iCCA^pps, which may provide insights into iCCAs with a different cell of origin.

## Methods

**Patient samples**. Fourteen patients had liver resection and were pathologically diagnosed as iCCA from January 2019 to January 2020 were enrolled for scRNA-seq. None of the patients received chemotherapy, radiotherapy, or any other anti-tumor therapy before surgery. Fresh paired tumor and non-tumor liver tissues were obtained during surgical resection. The adjacent normal tissues were at least 3 cm away from the matched tumor tissue. This study was conducted in accordance with the ethical standards of the Research Ethics Committee of Zhongshan Hospital with patients' informed consent. Written informed consent was obtained from all patients involved in this study for the use of their tissue samples and clinical information.

**Tissue microarray, immunohistochemistry, and Alcian blue staining**. Paraffin-embedded tissue samples from 201 iCCA patients who underwent primary and curative resection for their tumor in Liver Cancer Institute, Zhongshan Hospital of Fudan University (Shanghai, China) between 2012 and 2015 were selected. All these cases were pathologically diagnosed as iCCAs and were verified experimentally before[51]. The tissue microarrays were baked at 60 °C for 1 h, dewaxed in xylene, rehydrated through a gradient concentration, and blocked the endogenous peroxidase activity by 3% hydrogen peroxide. The sections were incubated with 10% goat serum for 30 min to block nonspecific binding sites and then incubated

with the primary antibodies including S100P (1:1500 dilution, ab133554, Abcam), SPP1 (1:2000 dilution, ab214050, Abcam), Hep-Par1 (1:2000 dilution, ab190706, Abcam), ARG1 (1:1000 dilution, ab133543, Abcam) and MUC5AC (1:1000 dilution, ab3649, Abcam) at 4 °C overnight. Detailed information on antibodies was provided in Supplementary Data 8. After repeated washing, the sections were incubated at room temperature with goat anti-mouse or goat anti-rabbit secondary antibody (Vector Lab, CA) and visualized by DAB solution and counterstained with hematoxylin. IHC staining score was assessed by two independent pathologists who were blinded to the patients' clinicopathological data. The score for IHC intensity was scaled as 0 for no IHC signal, 1 for weak, 2 for moderate, and 3 for strong. A positive IHC stain was defined by a visible staining pattern (score 1 to 3) compared to the negative control (score 0). Alcian blue staining was performed to evaluate mucin content using an Alcian blue staining kit (C0155M, Beyotime) following the manufacturer's instructions. The score for mucin content was scaled as 0 for accumulation of mucin within <10% of glandular lumens; 1 for accumulation of mucin within 10 to 50% of glandular lumens; and 2 for accumulation of mucin within >50% of glandular lumens or frequent intracytoplasmic mucin as previous study did[19].

**Preparation of single-cell suspensions**. Fresh iCCA tumor tissues and adjacent non-tumor liver tissues were obtained immediately following tumor resection and transferred to the 50 mL centrifugal tube filled with RPMI-1640 medium (Gibco) with 10% fetal bovine serum (Gibco) and transported rapidly to the laboratory on ice. Specimens were then washed twice with cold 1× PBS (Gibco) and digested with Miltenyi Tumor Dissociation Kit and the GentleMACS (Miltenyi, Bergisch Gladbach, Germany) following the manufacturer's instructions. The dissociated cells were subsequently passed through a 70 μm cell-strainer (BD) to remove clumps and undigested tissue. After centrifugation, the cell pellet was washed twice with MACS buffer (PBS containing 1% FBS, 0.5% EDTA, and 0.05% gentamycin) and then re-suspended in sorting buffer (PBS supplemented with 1% FBS). Single-cell suspensions were stained with DRAQ5 (1:200, 10 min, 4084, CST,) and DAPI (1:200, 5 min, 422801, Biolegend). Finally, DRAQ5 + DAPI- cells were sorted into RPMI-1640 media supplemented with 10% FBS by FACSAria (BD Biosciences).

**Single-cell RNA sequencing**. Libraries for scRNA-seq were generated using the Chromium Single Cell 3′ library and Gel Bead & Multiplex Kit from 10x Genomics. 10×Genomics Chromium barcoding system was used to construct a 10× barcoded cDNA library following the manufacturer's instructions. All libraries were sequenced on Illumina HiSeq 4000 until sufficient saturation was reached.

**scRNA-seq data processing**. CellRanger (v3.1.0) was applied for read mapping and gene expression quantification. Cells with less than 1000 UMIs or >20% mitochondria genes were excluded. We also used three algorithms (DoubletFinder, DoubletDetection, and Scrublet)[52–54] to find doublets and remove cells which were identified as a doublet by at least one algorithm. The total number of transcripts in each cell was normalized to 10,000, followed by log transformation. Then we used Seurat (v3)[55] to detect highly variable genes, perform PCA, graph-based clustering, and t-SNE.

**Classification of malignant cells**. As malignant cells harbor significantly more copy number variation (CNV) than normal cells, we estimated CNV from scRNA-seq following the steps described in the previous study[56] and made some minor improvements. In brief, we first restricted our target cells to epithelial cells defined by both SingleR. Then, genes were sorted according to their genomic location at each chromosome, and a sliding window of 100 genes was applied to calculate the average relative expression values to derive CNVi (CNV of the ith window). Epithelial cells from P02 and P04 (peripheral normal liver tissue) were used as a

reference in the above step. Next, we defined the CNV score of each cell as the mean of squared CNVi across all windows. In addition, we calculated the CNV correlation score by computing the Spearman correlation of the CNVi of a cell and the average CNVi of the single-cells with the top 3% CNV scores from the same tumor. Malignant cells were then defined as those with CNV signal above 0.04 and CNV correlation above 0.5.

**Classification of nonmalignant cells.** For all nonmalignant cells, we first used SingleR[57] to classify cells into seven major cell types: myeloid cell, NK cell, CD8+ T cell, CD4+ T cell, B cell, endothelial cell, and fibroblast. Other cell types (e.g., hepatocyte, neutrophil, mast cell, and normal epithelial cells) with fewer than 500 cells are excluded. Then we applied the graph-based clustering method implemented in Seurat to group cells into subtypes and each subtype was further annotated according to its marker genes.

**Bulk whole-exome sequencing and data processing.** DNA was extracted from iCCA tumor and non-tumor liver tissues from these fourteen patients using a DNeasy Blood and Tissue kit (Qiagen), and DNA concentration and purity were determined using a NanoQuant Plate Infinite M200 PRO reader (Tecan Austria GmbH). After enrichment of exonic DNA fragments with a SureSelect Human All Exon Kit (Agilent, 50 Mb V5), sequencing was performed on Illumina NovaSeq 6000.

Raw sequencing reads were mapped to human genome version 38 (hg38) using BWA-MEM[58]. After removing duplicated reads, SNV and indel were detected using Mutect2 (https://doi.org/10.1101/861054) and annotated with Oncotator[59]. Copy number alteration (CNA) was identified using FACETS[60].

**Tumor heterogeneity analysis.** For WES data, the cancer cell fraction (CCF) and clonality of each mutation was determined following the process described in Nicholas et al.[61] Genomic heterogeneity was calculated as the proportion of sub-clonal mutations in a tumor. For scRNA-seq data, we estimated transcriptomic heterogeneity according to the method in Ma et al[11].

**Differential expression and pathway analysis.** Differentially expressed genes (fold change >4 and P value < 0.001) were identified using the QLF model implemented in edgeR (v3.26.3)[62]. Pathway enrichment analysis was performed using clusterprofiler[63] based on GOBP gene sets from MSigDB.

**Gene regulatory network inference.** Gene regulatory networks were identified using SCENIC (v1.1.0)[24] with default settings. To reduce the computing time, a python implementation in SCENIC (GRNBoost) was used.

**Developmental trajectory analysis.** Monocle[64] was applied to infer the developmental trajectory with each tumor. Only the top 1000 variable genes identified by differentialGeneTest were selected for constructing the developmental tree.

**Dual-luciferase assay.** The dual reporter plasmid expressing firefly luciferase under the human S100P promoter and Renilla luciferase under the SV40 promoter was constructed. Different concentrations of expression plasmids were transiently transfected into the HEK-293T cells (purchased from ATCC) with Renilla luciferase plasmid. Firefly luciferase activity was measured with a Dual-Luciferase Assay Kit (Promega) 24 h after transfection and normalized with a Renilla luciferase reference plasmid. Results are assessed as the ratio of Firefly luciferase activity to Renilla luciferase activity.

**RNAi and transfection.** Human CREB3L1 siRNA (si-CREB3L1) lentivirus vectors and nonspecific siRNA (si-Ctrl) lentivirus vectors were synthesized by GeneChem Technology (Shanghai, China). The si-CREB3L1 sequences are at nucleotide positions131–149 (CGGAGAACATGGAGGACTT) as reported previously[65]. Non-targeting siRNA was used as the negative control. Lentivirus transfection was performed following the manufacturer's instructions and the efficiency of silencing was confirmed by immunoblotting.

**Transwell invasion assay.** Cell invasion was determined by Transwell invasion assay. Briefly, transwell inserts were firstly coated with Matrigel (BD, USA). Then, $1 \times 10^5$ HuCCT1 (purchased from Chinese Academy of Sciences Shanghai Branch Cell Bank, Shanghai, China) or RBE (purchased from Cell Resource Center of Tohoku University, Tohoku, Japan) cells suspended in 0.2 mL serum-free medium were added into inserts and 0.5 mL medium containing 20% FBS was added to the lower compartment as a chemoattractant. After culturing for 48 h, the cells on the upper membrane were carefully removed using a cotton bud, and cells on the lower surface were fixed with methanol for 15 min and successively stained with 0.1% crystal violet solution for 10 min. Photographs were then taken and the number of cells that passed through the Matrigel were counted. Assays were performed in duplicate in three independent experiments.

**Multiplex immunohistochemistry and quantitative analysis.** In brief, 4-μm FFPE TMAs sections were deparaffinized in xylene and then rehydrated in 100, 90, and 70% alcohol successively. Antigen unmasking was performed with a preheated epitope retrieval solution, endogenous peroxidase was inactivated by incubation in 3% $H_2O_2$ for 20 min. Next, the sections were pre-incubated with 10% normal goat serum and then incubated overnight with primary antibodies panel 1: CK19 (1:3500 dilution, ab52625, Abcam), S100P (1:3000 dilution, ab133554, Abcam), SPP1 (1:2000 dilution, ab214050, Abcam), CD68 (1:2000 dilution, 76437, CST), CCL18 (1:1000 dilution, ab104867, Abcam); panel 2: CK19 (1:3500 dilution, ab52625, Abcam), ID3 (1:2000 dilution, A5375, ABclonal), PDGFRβ (1:3000 dilution, ab32570, Abcam); panel 3: EPCAM (1:2000 dilution, ab223582, Abcam), S100P (1:3000 dilution, ab133554, Abcam), PSCA (1:2000 dilution, sc-80654, Santa Cruz Biotechnology); panel 4: CD45 (1:2500 dilution, ab40763, Abcam), CD3 (1:2000 dilution, ab16669, Abcam), CD68 (1:3000 dilution, ab213363, Abcam), CD206 (1:2000 dilution, 91992, CST), CD20 (1:2500 dilution, ab78237, Abcam), CD56 (1:2000 dilution, ab220360, Abcam); panel 5: CD4 (1:2000 dilution, ab133616, Abcam), CD8 (1:2500 dilution, ab237709, Abcam), PD1 (1:3000 dilution, ab52587, Abcam), FOXP3 (1:2000 dilution, ab215206, Abcam). Detailed information of antibodies was provided in the Supplementary Data 8). Next, sections were incubated with the corresponding HRP-conjugated goat anti-mouse or goat anti-rabbit second antibodies (Vector Lab, CA) for 30 min at room temperature. The antigenic binding sites were visualized using the OPAL dye. Opal −520 (PerkinElmer Inc.), Opal- 570 (PerkinElmer Inc.), Opal −620 (PerkinElmer Inc.), Opal -650 (PerkinElmer Inc.), Opal -690 (PerkinElmer Inc.) were applied to each antibody, respectively.

Data were analyzed as previously described[66]. Images were analyzed and quantified by inForm software (v2.3, PerkinElmer Inc.) based active machine learning algorithm with a pre-visual cutoff followed by single-cell based mean pixel fluorescence intensity to achieve accuracy. A threshold value of each marker was identified and displayed by both FCS Express 6 Plus v6.04.0034 (De Novo Software) with FACS alike density plot and Inform Score that could adjust the cutoff based on the score map and original staining images to improve the accuracy.

**Statistical analysis.** Statistical analysis was performed with the R (v3.6.1), SPSS (v22, IBM, Armonk, NY), and Prism 6.0 (SanDiego, CA) softwares. Comparisons were performed using $\chi^2$ test and unpaired two-sided Wilcoxon rank-sum test unless specified. The cumulative survival time was estimated by Kaplan–Meier estimator with a log-rank test.

**Reporting summary.** Further information on research design is available in the Nature Research Reporting Summary linked to this article.

## Data availability

The raw sequencing data reported in this paper (including scRNA-seq and WES data) has been deposited in the Genome Sequence Archive in National Genomics Data Center under the accession number HRA000863, which is accessible at. The raw sequencing data are available for non-commercial purposes under controlled access because of data privacy laws, and access can be obtained by request to the corresponding authors. The request will be passed within 1 week and then the users will be given a download link valid for 1 year to download the raw data. For public datasets analysis, Jusaka et al.'s dataset[20] (including 81 iCCAs and 34 ECCs) were retrieved from GSE89749 and GSE89803 and Job et al.'s dataset[21] (including 78 iCCAs) was retrieved from ArrayExpress with accession number E-MTAB-6389. Source data are provided with this paper. The remaining data were available within the Article, Supplementary Information, or Source Data file. Source data are provided with this paper.

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

## Acknowledgements

We thank the High-Performance Computing Platform of the Center for Life Sciences (Peking University) for providing the data analysis platform. The study was supported by project grants from the National Natural Science Foundation of China (Nos. 81961128025, 91859105, 11971039, and 31800743), Programs of the Science and Technology Commission of Shanghai (No. 22YF1407200, 20JC1418900 and 19XD1420700), the Strategic Priority Research Program (No. XDPB0303), Frontier Science Key Research Project (No. QYZDB-SSW-SMC036), Chinese Academy of Sciences, Sanming Project of Medicine in Shenzhen (No. SZSM202003009), and National Key Basic Research Project of China (No. 2020YFE0204000). This study was also supported by the Sino-Russian Mathematics Center.

## Author contributions

Q.G., R.X., X.Z., and J.F. contributed to study design and supervised the study. G.S. and Q.G. contributed to writing the manuscript. Y.S., S.H., Y.W., and J.-X.L. assisted in the data analysis. G.S., L.M., and J.M. performed immunohistochemical staining and image analysis. Ju.Z., Yp.L., J.L., and Sy.J. assisted in the preparation of the experiments. S.Y., D.R., Yf.C., and Ym.L. aided in the collection of tissue samples. S.J. and Xl.W. assisted with data collection. S.Z., A.K., Xy.W., and Y.J. assisted in histopathological analysis. Y.C., J.Z., and J.F. made intellectual contributions.

## Competing interests

The authors declare no competing interests.
