## [Peer Review File · Nature Communications]

Single-Cell Transcriptomic Analysis Suggests Two Molecularly Distinct Subtypes of Intrahepatic CholangiocarcinomaReviewers' Comments:

Reviewer #2:

Remarks to the Author:

The study by Song et al. utilized scRNA-seq to investigate two distinct subtypes of iCCA. While S100P is a well-known marker for iCCApHl (perihilar large duct type), the identification of SPP1 in defining iCCAppS (peripheral small duct type) is potentially useful. However, there are concerns regarding the robustness of classification of pHL and pPS subtypes of iCCA in different datasets. Besides, in the study by Ma et al. 2021 (Pubmed: 34216724), SPP1 expression was shown to be tightly associated with tumor cell evolution and microenvironmental reprogramming. It is important to compare and contrast the current findings with this study. Besides, the key message of the study is not strong.

1. How was the classification of iCCA cases into perihilar large duct (pHL) and peripheral small duct (pPS) subtypes? Was the classification confirmed by conventional methods other than examination of gene markers using scRNA-seq or RNA-seq? It is necessary to use different methods to confirm the pHL and pPS subtype status of the cases. The current definition based on only S100P and SPP1 expressions in scRNA-seq or RNA-seq is not convincing.
2. Were there any markers other than Hep-Par 1 used to determine hepatocyte contamination in the samples? Since hepatocyte-like cells were identified in the analysis, it is necessary to robustly rule out the chance of potential hepatocyte contamination in the tissue sampling.
3. What is meant by malignant and non-malignant cells? Are they referred to malignant cholangiocytes and non-malignant cholangiocytes? Or non-malignant cells are the immune and stromal cells?
4. How would you account for the concurrent expression of S100P and SPP1 in some cases as shown in Fig 2A and C?
5. What is the proportion of S100P+SPP1- and S100P-SPP+ cells in each patient?
6. In many of the figures, the cells were denoted as S100P+SPP1- and S100P-SPP+ cells. Since the expressions of S100P and SPP1 are likely patient-specific, it is necessary to show the proportion of the S100P+SPP1- and S100P-SPP+ cells in each patient and to determine whether the findings are restricted to patients or associated with S100P+SPP1- and S100P-SPP+ cells.
7. In figure 4E and F, what is the test used for comparing the inflammatory and M1/M2 scores? It seems the 2 groups look very similar based on the dot plot and the box plot. Besides, how would you explain SPP1+ macrophages are associated with both pro- and anti-inflammation? It is somehow contradictory.
8. What is the rationale for comparing ID3 expression between ALB+ and ALB- cells? Obviously, there should be other genes also having significantly different expression between groups of cells, why did you choose ID3 for studying?
9. How are the findings of the current study as compared to the study by Ma et al. 2021? What is the novelty of the current study?
10. Most of the findings of the study are exploratory and based on association. It is necessary to identify and confirm key findings based on the scRNA-seq discovery e.g. the functional role of CREB3L1 or biological difference between SPP1+CCL18- and SPP1-CCL18+ macrophages. They are likely the major novelties of the study.

Reviewer #3:

Remarks to the Author:

The authors studied tumor and peritumor tissue samples from a cohort of 14 patients with intrahepatic cholangiocarcinoma. WES and single cell RNA-seq was performed. Data was verified in a TMA cohort and two publically available cohorts. Additional studies included IHC analysis for immune cells in a set of 68 patients.

This study adds to prior studies mentioned by the author describing results on single cell RNA-seq of cholangiocarcinoma samples by Zhang et al published in JHep in 2020 and Ma et al, published in Cancer Cell in 2019. The authors missed a very recent follow-up study by Ma et al, published in J.Hep 2021, which is quite relevant to the findings described here since it also studied SPP1. Overall, I very much like the study because it is timely, provides interesting data and appears to be well conducted. The sample size is good. My major concern is the lack of clinical relevance (survival data would be interesting, but may be too early). Also, from a clinical perspective I am not aware that the two different types of iCCA differ significantly. Unfortunately, the study is rather descriptive and almost no functional data is provided. It would be so interesting to go a step further and try to better understand how the described findings will lead to a specific change in the tumor microenvironment and thereby affect outcome and or prognosis, but the authors decided to stay at a rather superficial level.

Reviewer #4:

Remarks to the Author:

This study examined scRNA data of ICC, which has been recognized as heterogenous tumor. The data of scRNA is very valuable to understand such a heterogenous tumors and they indicated the heterogeneity of cell origin, differentiation, and tumor microenvironment.

1) They say that tumors were negative for Hep-Par 1 (a sensitive marker of hepatocytes) expression, but they analyzed ALB expression and discussed hepatocyte-like differentiation. And usually, normal hepatocyte cells are contaminated in bulk tissues.

2) They identified 23,667 malignant cells by inferring large-scale copy number variations (CNVs) (Fig. 1c and Supplementary Fig. 1c). This issue is very weird and they should describe more in Results. This CNV prediction is precise?

3) It is interesting that the higher percentage of HBsAg positive status and liver cirrhosis in S100P SPP1+ iCCAs further support the notion that iCCAs usually develops on a background of chronic liver disease

Fujimoto A et al (Nat Commun 6: 6120, 2015) and Wardel CP et al. (J Hepatol 36:959-969, 2018) demonstrated that some of ICCs, which had a background of chronic liver disease showed quite different molecular profiles and were predicted to be derived from different cell origins.

The associations with HB infection and chronic hepatitis should be emphasized and discussed more. Did they have significant association with chronic hepatitis (pathological validated), not liver cirrhosis?

4) I would like to make sure that SPP1+ macrophages expressed SPR1 gene as well as malignant ICC cells? If so, it is very complicated and it should be described more detail to avoid misunderstanding.

5) They should show tSNE blot for other ICC-related markers, such as KRT19, EMA and NESTIN Especially, NESTIN is a stemness marker in combined type cHCC/CC, which is also heterogenous and can be similar to small duct cell type of ICC in some aspects.

6) iCCApps may have more actionable mutations and treatable by new molecular drugs such as IDH1 mutations and FGFR fusions. Did they detect FGFR fusions in these 14 ICCs?

7) Regarding to immune cell expression, they should examine T cell and T cell exhaustion markers such as PD1 as well as Macrophage, because primarily T cells are working as immune editing in cancer and it would be more interesting if they find any difference between T cells in these two type of ICCs.

Response to the reviewer's comments:

We would like to thank the reviewers for their helpful comments and critiques. We list our responses below.

Reviewer #2:

The study by Song et al. utilized scRNA-seq to investigate two distinct subtypes of iCCA. While S100P is a well-known marker for iCCAp_{hl} (perihilar large duct type), the identification of SPP1 in defining iCCAp_{ps} (peripheral small duct type) is potentially useful. However, there are concerns regarding the robustness of classification of p_{hl} and p_{ps} subtypes of iCCA in different datasets. Besides, in the study by Ma et al. 2021 (Pubmed: 34216724), SPP1 expression was shown to be tightly associated with tumor cell evolution and microenvironmental reprogramming. It is important to compare and contrast the current findings with this study. Besides, the key message of the study is not strong.

1. How was the classification of iCCA cases into perihilar large duct (p_{hl}) and peripheral small duct (p_{ps}) subtypes? Was the classification confirmed by conventional methods other than examination of gene markers using scRNA-seq or RNA-seq? It is necessary to using different methods to confirm the p_{hl} and p_{ps} subtype status of the cases. The current definition based on only S100P and SPP1 expressions in scRNA-seq or RNA-seq is not convincing.

Reply: Thank you for your comments and suggestions. Though the HE staining could be used to visualize bile duct structures, 32.4% of iCCA could not be accurately classified as iCCA^{p_{hl}} or iCCA^{p_{ps}} according to histological morphology¹. That is because these iCCAs contained mixtures of typical large duct and small duct types and also displayed atypical histology, such as columnar cell tumors without mucin, cuboidal cell tumors with abundant mucin production, or poorly differentiated tumors. It is also very difficult to distinguish

them by assessing the location of the tumors because some of iCCAs are larger than 5 cm in size^{1, 2, 3}. The combined detection of the expression of multiple tissue markers has higher accuracy for iCCA classification, and previous studies have found that S100P is an effective distinguishing marker for iCCA^{phl} with high discriminative value^{1, 4, 5}. We extended this line and proposed a new classification system based on S100P and SPP1. To substantiate the newly defined subtypes of the iCCAs, we combined the staining of S100P and SPP1, together with the Alcian blue staining (to detect the mucus secreted by mucous tumor cells), immunohistochemical staining of MUC5AC (essential for mucus production), and HE staining (morphology) in 201 TMA samples (Figure R1 and a new Supplementary Fig. 4). We totally identified 68 S100P+SPP1-, 118 S100P-SPP1+, 12 S100P-SPP1-, and 3 S100P+SPP1+ iCCAs in this cohort. Our results showed that all these 68 (33.83%) S100P+SPP1- iCCAs were positive for MUC5AC and mucin production. Among the 118 (58.71%) S100P-SPP1+ iCCAs, only 6 cases were positive for MUC5AC and 3 cases were positive for mucin production. Among the 12 (5.97%) S100P-SPP1- iCCAs, 3 cases were positive for MUC5AC and 3 for mucin production, respectively. For these 3 (1.49%) S100P+SPP1+ iCCAs, 2 cases were positive for MUC5AC and all the 3 were positive for mucin production. Thus, our results demonstrated that these S100P+SPP1- iCCAs were more like iCCA^{phl} (all were positive for MUC5AC and mucin production), while S100P-SPP1+ iCCAs were more like iCCA^{pps} (mostly were negative for MUC5AC and mucin production). There were indeed a small number of S100P+SPP1+ (3 cases, 1.49%) and S100P-SPP1- (12 cases, 5.97%) iCCAs present in our TMA cohort. We found that most of S100P-SPP1- iCCAs showed the characteristics of iCCA^{pps} (rarely MUC5AC and mucin production), while S100P+SPP1+ showed the characteristics of iCCA^{phl} (all positive for mucin production). However, due to their small sample size and their low proportion in the iCCA, we did not pay too much attention to them and further studies in larger cohorts are required to validate these findings. The

same experiments were also performed with our 14 scRNA-seq iCCA cases, and results showed that seven S100P+ iCCAs (P02, P03, P04, P06, P16, P17, and P18) were all positive for MUC5AC and mucin production, while the other seven S100P- iCCAs (P09, P10, P12, P13, P14, P15, and P19) were negative for this (data not shown).

Overall, although S100P is a recognized marker of iCCA^{phl}, SPP1 as a marker of iCCA^{pps} has not reported before. We here for the first time demonstrated that SPP1 was highly expressed in iCCA^{pps} and 92.5% iCCAs can be classified as iCCA^{phl} and iCCA^{pps} according to the expression of S100P and SPP1. The findings of the present study are important in the light of clinical application and directions for future research. We have supplemented this part of the results in the revised manuscript (Supplementary Fig. 4).

Figure R1 Representative immunohistochemical staining, mucin production, and histology (hematoxylin and eosin) of iCCA (n=201).

Typical large duct type refers to iCCAs composed of columnar cells with abundant mucin production. Typical small duct type refers to iCCAs composed of cuboidal or low columnar cells with no or rare mucin production.

2. Were there any markers other than Hep-Par 1 used to determine hepatocyte contamination in the samples? Since hepatocyte-like cells were identified in the analysis, it is necessary to robustly rule out the chance of potential hepatocyte contamination in the tissue sampling.

Reply: Thanks for the comments. In addition to Hep-Par 1, arginase-1 (Arg-1) was proved to be specifically expressed in hepatocellular carcinoma (HCC) but not in iCCA^{1, 6}. We performed immunohistochemical staining on the 14 scRNA-seq samples and found that all 14 cases were negative for Hep-Par1 and Arg-1 expression, indicating their non-HCC origin (Figure R2a and a new Supplementary Fig. 1b). This result was also verified at the single cell level (Figure R2b). We are confident that there is no hepatocyte contamination in our tumor samples based on the following aspects. First, before performing scRNA-seq of iCCA tumor tissues, we completely trimmed the normal liver tissues around the tumor and washed the tissues repeatedly. Library for iCCA and adjacent non-tumor liver tissues were constructed and sequenced individually and would not be mixed. Second, we also identified ALB+ tumor cells in the data from Ma *et al*⁷. Similar to the results from our scRNA-seq data, these ALB+ tumor cells were mainly derived from S100P-SPP1+ iCCA samples with high expression of other hepatocyte specific genes such as CPB2, ASGR1, and FGA, which further illustrates the existence of hepatocyte-like tumor cells in the iCCA^{pps} (Figure R2c-f). Third, we found that ALB+ tumor cells had comparable CNV scores with ALB- tumor cells, as

compared to normal epithelial cells, and both groups had significant higher CNV scores. This result indicated that ALB+ cells also contained considerable CNV events, supporting their malignant characteristics (Figure R2g).

Figure R2 The identification of ALB+ tumor cells in iCCA samples. a Representative immunohistochemical staining of Hep-Par1 and Arg-1 in P09 scRNA-seq iCCA case. **b** t-SNE plot showing expression of Arg-1 in our scRNA-seq data. **c** t-SNE plot of all malignant cells coloured by sample origin from Ma *et al.* data. **d** t-SNE plot displaying the expression of S100P and SPP1 in all the malignant cells from Ma *et al.* data. **e** t-SNE plot showing expression of ALB. **f** Heatmap displaying expression levels of hepatocyte related genes from Ma *et al.* data. **g** CNV score of ALB+/ALB- malignant cells and normal epithelial cells.

3. What is meant by malignant and non-malignant cells? Are they referred to malignant cholangiocytes and non-malignant cholangiocytes? Or non-malignant cells are the immune and stromal cells?

Reply: Sorry for the confusion. In our original manuscript, malignant cells referred to tumor cells identified by inferring large-scale copy number variations (CNVs), while non-malignant cells were referred to all other cells, including all immune cells, stromal cells and normal cholangiocytes. Note that some small subsets of cells, including hepatocytes, neutrophils, mast cells and normal epithelial cells, with fewer than 500 cells were excluded from our analysis. We have modified this in the revised manuscript. (Page 8, Line 169-171)

4. How would you account for the concurrent expression of S100P and SPP1 in some cases as shown in Fig 2A and C?

Reply: Thanks for the comments. Our scRNA-seq results showed that P03, P04 and P06 iCCAs had a higher proportion of S100P+SPP1+ cells (80.4%, 41.6%, and 49.5% respectively, Supplementary Fig. 2f). However, due to the relatively small number of tumor cells detected in these three samples (455,

478, and 293 respectively), the proportion of S100P+SPP1+ cells in all the tumor cells from the 14 scRNA-seq samples is very low (5.98%, Supplementary Fig. 2e). Similar results were observed in Ma *et al.*'s data that though C60 case contained high proportion of S100P+SPP1+ cells (91.1%), these cells account for a very low proportion of the total cells (5.62%, Figure R3a, b). We speculate that these cells are a type of tumor cells that present in iCCA^{phl}. To validate this hypothesis, we performed t-SNE on P04T and P06T, the only two samples with enough S100P+SPP1+, S100P+SPP1- and S100P-SPP1+ cells. According to the dimension reduction result, the global expression profile of S100P+SPP1+ showed a higher degree of similarity to S100P+SPP1- than the S100P-SPP1+ cells (Figure R3c and a new Supplementary Fig. 3e). Furthermore, immunohistochemistry on iCCA tumor slides revealed that S100P+SPP1+ cells existed in iCCA^{phl}. We further performed S100P and SPP1 staining on another 17 iCCA cases and found that 16 of them can be clearly defined as S100P+SPP1- iCCA or S100P-SPP1+ iCCA by S100P and SPP1 expression, and only one case was S100P-SPP1- iCCA. We found that the S100P+SPP1+ double positive cells were mostly present in the invasive regions of cancer nodules in certain iCCA^{phl} cases (Figure R3d and a new Supplementary Fig. 3f). Accumulating evidence has revealed SPP1 acts as a significant mediator of modulating tumor invasion and metastasis⁸, implying these double positive cells may be involved in the progression of iCCA. We have added these results to the revised manuscript (Supplementary Fig. 3).

Figure R3 The presence of S100P+SPP1+ tumor cells in iCCA. a Graph of the proportions of four subtypes of iCCA cells in ten iCCA patients from Ma *et al.*'s dataset. **b** The pie chart shows the percentage of the four groups of tumor cells in the total tumor cells from Ma *et al.*'s dataset. **c** t-SNE plots of four subtypes of iCCA tumor cells from P04 and P06 patients. **d** Immunohistochemical staining for S100P and SPP1. Arrows refer to SPP1+ tumor cells.

5. What is the proportion of S100P+SPP1- and S100P-SPP+ cells in each patient?

Reply: Proportions of these cells in different patients are shown in Figure R4 and a new Supplementary Fig. 2f.

Figure R4 Stacked bar plot of the proportions of four subtypes of iCCA tumor cells in fourteen iCCA patients.

6. In many of the figures, the cells were denoted as S100P+SPP1- and S100P-SPP+ cells. Since the expressions of S100P and SPP1 are likely patient-specific, it is necessary to show the proportion of the S100P+SPP1- and S100P-SPP+ cells in each patient and to determine whether the findings are restricted to patients or associated with S100P+SPP1- and S100P-SPP+ cells.

Reply: Following the reviewer's suggestion, we have added the proportion of the S100P+SPP1- and S100P-SPP+ cells in each patient in our revised manuscript (Supplementary Fig.2f). S100P+SPP1- cells constituted the majority of tumor cells in P02, P16, P17, and P18 iCCA^{phl} cases, while S100P-SPP1+ constituted the majority of tumor cells in P09, P10, P12, P13, P14, and P15 iCCA^{pps} cases (Figure R5a, b). We identified a higher proportion of S100P+SPP1+ cells from P03, P04, and P06 iCCA^{phl} cases partly due to the

heterogeneity of the studied samples as we recovered in Figure R3d. These results were very similar to those of Ma *et al.*, in which the expressions of S100P and SPP1 also showed a patient-specific characteristic (Figure R5c, d). To avoid patient bias, all the S100P+SPP1- and S100P-SPP+ cells derived from different cases were analyzed all together.

Figure R5 Four subtypes of iCCA tumor cells identified by scRNA-seq. a The pie chart shows the percentage of the four subtypes of tumor cells in the total tumor cells from our scRNA-seq data. **b** Graph of the proportions of four subtypes of iCCA tumor cells in fourteen iCCA patients from our scRNA-seq data. **c** Graph of the proportions of four subtypes of iCCA cells in ten iCCA patients from Ma *et al.*'s dataset. **d** The pie chart shows the percentage of the four groups of tumor cells in the total tumor cells from Ma *et al.*'s dataset.

7. In figure 4E and F, what is the test used for comparing the inflammatory and M1/M2 scores? It seems the 2 groups looks very similar based on the dot plot and the box plot. Besides, how would you

explain SPP1+ macrophages are associated with both pro- and anti-inflammation? It is somehow contradictory.

Reply: To estimate pro-/anti-inflammatory ability for macrophages, we applied a scoring method widely used in single cell studies using the gene sets given in Azizi *et al* (2018)⁹. The difference between two groups were assessed by the non-parametric Wilcoxon rank-sum test. The medians of these two groups are similar because the dynamic range of this scoring method is small. However, as the number (7,463 cells) is large, it still reaches statistical significance.

As shown in Figure R6, SPP1+ macrophages showed higher expression of pro-inflammatory genes (IL1B, CCL3, CCL5, and IL1R2) and anti-inflammatory genes (TNFRSF1B, IL10, TGFB1, and IL1R2) than CCL18+ macrophages. The seemingly contradictory results may reflect the real activation status of macrophages and the advantages of scRNA-seq. By this technology, we can see new macrophage subsets and obtain more of their biological functions than bulk RNA sequencing. Also, it has been reported that macrophage activation in tumor microenvironment did not follow the classical polarization pattern^{9, 10}. Here, our results revealed the diversity of macrophages in regulating tumor immunity and that some of them did not only have a simple anti-inflammatory or pro-inflammatory ability. Furthermore, the expression of anti-inflammatory molecules such as IL10 is likely the negative feedback of the pro-inflammatory response by activated macrophages.

Figure R6 Boxplots showing the expression of pro/anti-inflammatory genes in Macro_c1_SPP1 and Macro_c2_CCL18.

8. What is the rationale for comparing ID3 expression between ALB+ and ALB- cells? Obviously, there should be other genes also having significantly different expression between groups of cells, why did you choose ID3 for studying?

Reply: Thanks for the comments. We chose ID3 as the marker of ALB- cells based on the following aspects. First, differential gene expression analysis between ALB+ and ALB- cells revealed that ID3 was highly expressed in ALB- cells (logFC: 2.05, $P < 0.01$) and its expression was mutually exclusive with ALB. Second, previous study has identified ID3+ cells at the early stages of development in human and mouse fetal livers, which were able to differentiate into both hepatocytes or cholangiocytes¹¹. We found these ID3+ tumor cells also highly expressed LGR5, a stem cell marker, suggesting their stemness

and differentiation potential. Third, it had also been reported that ID3 could promote stem cell features and predict chemotherapeutic response of iCCA¹², further demonstrated that these cells might play an important role in the occurrence and progression of iCCA.

9. How are the findings of the current study as compared to the study by Ma et al. 2021? What is the novelty of the current study?

Reply: Thanks for the comments. The purpose and finding of the previous study⁷ (Ma *et al.* 2021) are totally different from those of our present study. In the study of Ma *et al.*, they mainly focused on tumor cell functional clonality and tumor evolution in response to treatment in HCC and iCCA. Though they identified SPP1 as a potential player in tumor cell evolution, it was still unknown about its role in the classification of iCCA. Compared with prior study, our innovation is mainly in the following aspects. 1) Two molecularly and clinically distinct types of iCCAs (S100P+SPP1- and S100P-SPP1+) were identified based on our scRNA-seq data. We proposed a promising diagnostic marker SPP1 for S100P-SPP1+ iCCA^{pps}. 2) We revealed the differences in tumor microenvironment and immune cell infiltration between these two iCCA subtypes. S100P+SPP1- iCCA^{phl} has significantly reduced levels of infiltrating CD3+ T cells, CD56+ NK cells, and increased CCL18+ macrophages compared to S100P-SPP1+ iCCA^{pps}. 3) S100P-SPP1+ iCCA^{pps} contains tumor cells at different status of differentiation, such as ALB+ hepatocyte differentiated cells and ID3+ stem cells, which may bring new druggable targets and novel possibilities for iCCA therapy. 4) We for the first time identified CREB3L1 as the transcription factor regulating S100P and showed that CREB3L1 plays an important role in promoting tumor cell invasion, which may be a potential target gene for iCCA^{phl} treatment. Collectively, our study brings new insights into iCCA subtypes and provides a theoretical basis for its precision therapy.

10. Most of the findings of the study are exploratory and based on association. It is necessary to identify and confirm key findings based on the scRNA-seq discovery e.g. the functional role of CREB3L1 or biological difference between SPP1+CCL18- and SPP1-CCL18+ macrophages. They are likely the major novelties of the study.

Reply: Thanks for suggestions. We understand that discovering the functional role of CREB3L1 or biological difference between SPP1+CCL18- and SPP1-CCL18+ macrophages may better reveal the mechanism of iCCA initiation and development. In this paper, we confirmed CREB3L1 as a transcription factor for S100P by SCENIC analysis and dual-luciferase report assay (Figure 3c-e). We also determined the role of CREB3L1 in tumor invasion by functional experiments (Figure 3f, g).

By unsupervised clustering, we identified two clusters of macrophages, Macro_c1_SPP1 (53.8% of total macrophages) and Macro_c2_CCL18 (46.2% of total macrophages), were significantly enriched in tumors compared with paired non-tumor tissues in iCCA (Figure 4a, b and Supplementary Fig. 7d). The Results from immunostaining further revealed the correlation between the infiltration pattern of these macrophage subsets and two iCCA subtypes (Figure 4g, h). As a subset of macrophages, SPP1+ macrophages had been identified in a variety of tumor types and found to be associated with tumor angiogenesis in colorectal cancer^{13,14,15}. By our scRNA-seq analysis, we found that Macro_c2_CCL18 macrophages showed a dominant M2-like phenotype, with elevated cytokine-cytokine receptor interaction, nitrogen and riboflavin metabolism compared to Macro_c1_SPP1 macrophages, but we did not conduct an in-depth study of the molecular mechanisms between these two subsets. In the present study, we mainly focused on the classification of iCCA and revealed the complexity and diversity of its tumor microenvironment. Our data provides a foundation upon which these future studies can be built and

also provides many potential directions for next steps with this research. Our group has been considering future study to further address these issues. Thank you again for your kind advise.

Reviewer #3:

The authors studied tumor and peritumor tissue samples from a cohort of 14 patients with intrahepatic cholangiocarcinoma. WES and single cell RNA-seq was performed. Data was verified in a TMA cohort and two publically available cohorts. Additional studies included IHC analysis for immune cells in a set of 68 patients.

This study adds to prior studies mentioned by the author describing results on single cell RNA-seq of cholangiocarcinoma samples by Zhang et al published in JHep in 2020 and Ma et al, published in Cancer Cell in 2019. The authors missed a very recent follow-up study by Ma et al, published in J.Hep 2021, which is quite relevant to the findings described here since it also studied SPP1. Overall, I very much like the study because it is timely, provides interesting data and appears to be well conducted. The sample size is good. My major concern is the lack of clinical relevance (survival data would be interesting, but may be too early). Also, from a clinical perspective I am not aware that the two different types of iCCA differ significantly. Unfortunately, the study is rather descriptive and almost no functional data is provided. It would be so interesting to go a step further and try to better understand how the described findings will lead to a specific change in the tumor microenvironment and thereby affect outcome and or prognosis, but the authors decided to stay at a rather superficial level.

Reply: We appreciate very much for the reviewer's comments. In the study of Ma et al., they mainly focused on tumor cell functional clonality and tumor

evolution in response to treatment in HCC and iCCA. Though they identified SPP1 as a potential player in tumor cell evolution, it was still unknown about its role in the classification of iCCA. In our study, we mainly focused on the molecular classification of this tumor and revealed the complexity and diversity of its tumor microenvironment. Compared with their study⁷ (Ma *et al.* 2021), our innovation is mainly reflected in the following aspects. 1) Two molecularly and clinically distinct types of iCCAs (S100P+SPP1- and S100P-SPP1+) were identified based on our scRNA-seq data. We proposed that SPP1 is a promising diagnostic marker for S100P-SPP1+ iCCA^{pps}. 2) We revealed the differences in tumor microenvironment and immune cell infiltration between these two iCCA types. S100P+SPP1- iCCA^{phl} has significantly reduced levels of infiltrating CD3+ T cells, CD56+ NK cells, and increased CCL18+ macrophages compared to S100P-SPP1+ iCCA^{pps}. 3) S100P-SPP1+ iCCA^{pps} harbors tumor cells at different status of differentiation, such as ALB+ hepatocyte differentiated cells and ID3+ stem cells, which may bring new druggable targets and novel possibilities for iCCA therapy. 4) We for the first time identified CREB3L1 as the transcript of S100P and playing important role in promoting tumor cell invasion, which may be a potential target gene for iCCA^{phl} treatment. Collectively, our data provides a foundation upon which these future studies can be built and also provides many potential directions for next steps with this research.

Reviewer #4:

This study examined scRNA data of ICC, which has been recognized as heterogenous tumor. The data of scRNA is very valuable to understand such a heterogenous tumors and they indicated the heterogeneity of cell origin, differentiation, and tumor microenvironment.

1. They say that tumors were negative for Hep-Par 1 (a sensitive marker of hepatocytes) expression, but they analyzed ALB expression and

discussed hepatocyte-like differentiation. And usually, normal hepatocyte cells are contaminated in bulk tissues.

Reply: Thanks for the comments. In addition to Hep-Par 1, arginase-1 (Arg-1) was proved to be specifically expressed in hepatocellular carcinoma (HCC) but not in iCCA. We performed immunohistochemical staining on 14 scRNA-seq samples and found that all these samples were negative for Hep-Par 1 and Arg-1 expression, indicating their non-HCC origin (Figure R7a and a new Supplementary Fig. 1b). This was also verified at the single cell level (Figure R7b). We are confident that there is no hepatocyte contamination in our tumor samples based on the following aspects. First, before we performed scRNA-seq of iCCA tumor tissues, we completely trimmed the normal liver tissues around the tumor and washed the tissues repeatedly. Library for iCCA tumor and adjacent non-tumor tissues were constructed and sequenced individually and wouldn't be mixed. Second, we also identified ALB+ tumor cells in the data from Ma *et al*⁷. Similar to the result from our data, these cells were mainly distributed in S100P-SPP1+ iCCA samples with high expression of other hepatocyte specific genes such as CPB2, ASGR1, and FGA, which further illustrated the existence of hepatocytes differentiated tumor cells in the iCCA^{pps} (Figure R7c-f). Third, we found that ALB+ cells had comparable CNV scores with ALB- cells, yet compared to normal epithelial cells, both groups had significant higher CNV scores. This indicated that the ALB+ cells also contained considerable CNV events, which validated their malignant characteristics (Figure R7g).

Figure R7 The identification of ALB+ tumor cells in iCCA samples. a Representative immunohistochemical staining of Hep-Par1 and Arg-1 in P09 scRNA-seq iCCA case. **b** t-SNE plot showing expression of Arg-1 in our scRNA-seq data. **c** t-SNE plot of all malignant cells coloured by sample origin

from Ma *et al.* data. **d** t-SNE plot displaying the expression of S100P and SPP1 in all the malignant cells from Ma *et al.* data. **e** t-SNE plot showing expression of ALB. **f** Heatmap displaying expression levels of hepatocyte related genes from Ma *et al.* data. **g** CNV score of ALB+/ALB- malignant cells and normal epithelial cells.

2. They identified 23,667 malignant cells by inferring large-scale copy number variations (CNVs) (Fig. 1c and Supplementary Fig. 1c). This issue is very weird and they should describe more in Results. This CNV prediction is precise?

Reply: We followed the process used in many single cell RNA-seq papers to identify malignant cells^{7,16,17,18,19}. There are two basic hypotheses for this analysis:

1) Large-scale CNV could be reflected by transcriptome profile, that is, amplification leads to high expression of genes while deletion leads to low expression of genes.

2) In contrast to normal cells, most malignant cells harbor CNVs.

Our CNV estimation process is detailed below. First, as malignant cells should originate from epithelial cells, we only considered epithelial cells (annotated by SingleR) for CNV analysis. Then we sorted all genes according to their genomic location at each chromosome, and applied a sliding window of 100 genes to calculate the average expression of each window i . This step is to eliminate gene-specific expression pattern. At the same time, we trimmed the expression values to $[-3,3]$ to avoid the influence of genes with extreme high/low expression. Then for each cell, using the average expression profile of normal epithelial cells from P02 and P04 (peripheral normal liver tissue) as a reference, we calculated the relative copy number ratio CNV_i for each window. Finally, two parameters for each cell were derived based on CNV_i :

1) CNV score: the sum of squared CNV_i across all windows.

2) CNV correlation score: For each sample we first identified the single cells whose CNV scores are ranked the top 3% among all single cells of the sample. We then calculate the average of CNV_i 's of the top-3% cells in each window and correlate the CNV_i of each single cell with the average CNV_i . The obtained correlation is the CNV correlation score.

To further validate the fidelity of our CNV estimation result, we compare our result with the CNV profiles estimated by whole exome sequencing from a large Chinese iCCA cohort²⁰ (Zou S, *et al.*, 2014). We found that the most frequent amplification peaks such as chr1q, chr7 and chr8, deletion peaks like chr1p, chr4, chr12q were also identified in our single cell data (Figure R8).

We have to admit that this approach has some limitations. For example, some cancer types are not driven by CNV, so identifying malignant cells by CNV will miss some malignant cells (false negative). However, here we believe that false negatives will not be too much because previous studies indicated that CNV occurs in most biliary tract^{20, 21, 22}. Until now, identifying malignant cells by transcriptome-inferred CNV is still the most reliable and widely used method so that most single cell studies utilized this method to extract malignant cells.

Figure R8 Copy number variations (CNVs) identified in iCCA. **a** Heat map shows large-scale CNVs for individual cells (rows) inferred based on the average expression of 100 genes surrounding each chromosomal position (columns). Red: amplifications; Blue: deletions. **b** CNV profiles estimated by whole exome sequencing from a large Chinese iCCA cohort (Zou S, *et al.*, 2014).

3. It is interesting that the higher percentage of HBsAg positive status and liver cirrhosis in S100P SPP1+ iCCAs further support the notion that iCCAs usually develops on a background of chronic liver disease Fujimoto A et al (Nat Commun 6: 6120, 2015) and Wardel CP et al. (J Hepatol 36:959-969, 2018) demonstrated that some of ICCs, which had a background of chronic liver disease showed quite different molecular profiles and were predicted to be derived from different cell origins. The associations with HB infection and chronic hepatitis should be emphasized and discussed more. Did they have significant association with chronic hepatitis (pathological validated), not liver cirrhosis?

Reply: Thanks for this valuable suggestion. In our TMA cohort (201 iCCA patients), 45 patients were HBsAg positive and 39 patients had liver cirrhosis. We here assessed their chronic hepatitis status of the included patients in the TMA cohort using the Batts-Ludwig scoring system. Statistical results showed that there were significantly higher percentages of chronic viral hepatitis patients in S100P-SPP1+ iCCA^{pps} than in S100P+SPP1- iCCA^{phl} (52.5% versus 29.4%, $P = 0.002$). It has been noted that the occurrence of these two subtypes of iCCA was related to different pathogenic factors^{3,4,23,24}. The iCCA^{pps} usually developed on a background of chronic viral hepatitis or liver cirrhosis compared with iCCA^{phl}, which often developed under primary sclerosing cholangitis (PSC) or liver fluke infection status²³. Our results

revealed that there were significantly higher percentages of HBsAg positivity, chronic viral hepatitis and liver cirrhosis in S100P-SPP1+ iCCA^{pps} than in S100P+SPP1- iCCA^{phl}, further highlighting their distinct pathogenic background. In addition, all the patients with liver cirrhosis we included were HBV related, so it was very difficult to compare the difference between chronic hepatitis B and liver cirrhosis in the involvement of iCCA. We have added these results in our revised manuscript (Supplementary table 3) and discussed this in the Discussion section. (Page 23, Lines 491-498)

4. I would like to make sure that SPP1+ macrophages expressed SPR1 gene as well as malignant ICC cells? If so, it is very complicated and it should be described more detail to avoid misunderstanding.

Reply: Thanks for the reviewer's expert suggestion. As a subset of macrophages, SPP1+ macrophages had been detected in local microenvironment of a variety of tumors, like HCC, breast cancer, and colorectal cancer^{13,14,25}. In this study, we confirmed the presence of these cells in iCCA tumor microenvironment and they are more enriched in SPP1+S100P- iCCA^{pps}. However, we cannot determine whether the form of SPP1 expressed by SPP1+ macrophages is the same as that of SPP1+ tumor cells, because two isoforms of SPP1 (also called osteopontin, OPN): a secreted form (sOPN) and an intracellular form (iOPN) has been identified with distinct functions²⁶ (Figure R9). These two forms of SPP1 are produced by translation from different alternative initiation sites that cannot be distinguished by scRNA-seq or immunohistochemistry. We realize that this is indeed a very complicated question, and to avoid confusion, we address this in the Discussion section of the revised manuscript. (Page 24, Lines 510-514)

Figure R9 Two isoforms of SPP1 generated by alternative translation. (Inoue M, *et al.*, 2011)

5. They should show t-SNE blot for other ICC-related markers, such as KRT19, EMA and NESTIN

Especially, NESTIN is a stemness marker in combined type cHCC/CC, which is also heterogenous and can be similar to small duct cell type of ICC in some aspects.

Reply: Thanks for this suggestion. We added these three t-SNE plots for KRT19, EMA and NESTIN in our revised manuscript (Figure R10 and new Supplementary Fig.1c, 2a). It has been recognized that both cHCC/CC and iCCA^{pps} can be originated from hepatic stem or progenitor cells²⁷. As a maker of bipotent progenitor oval cells, NESTIN is greatly increased in cHCC/CC and has been proposed as a possible diagnostic biomarker²⁸. Here, the results of our scRNA-seq showing that NESTIN was mostly expressed in SPP1+S100P-iCCA^{pps}, suggesting the possible similarities between cHCC/CC and iCCA^{pps}, but further studies are required to verify this hypothesis.

Figure R10 t-SNE plots showing the expression of iCCA-related markers.

6. iCCApps may have more actionable mutations and treatable by new molecular drugs such as IDH1 mutations and FGFR fusions. Did they detect FGFR fusions in these 14 ICCs?

Reply: We fully agree with the reviewer's comments. FGFR2 fusions are important events in iCCAs and commonly found in 10–15% of cases. Since fusion genes cannot be accurately detected by whole-exome sequencing (WES) or single cell RNA-seq (scRNA-seq), we here further performed fluorescence in situ hybridization (FISH) to detect FGFR2 fusions in these 14 iCCAs. Only one sample (P09), derived from the S100P-SPP1+ iCCA^{pps}, exhibited a positive signal for the FGFR2 gene fusion (Figure R11). We also identified one IDH1 mutation case in S100P-SPP1+ iCCA^{pps} (Supplementary Fig.5c). However, due to the small number of cases with IDH1 mutation and FGFR fusion in our data, we did not conduct an in-depth analysis this aspect.

Figure R11 FGFR2 fusions validated by FISH using two specific probes for the 5' (red) and 3' (green) part of FGFR2.

7. Regarding to immune cell expression, they should examine T cell and T cell exhaustion markers such as PD1 as well as Macrophage, because primarily T cells are working as immune editing in cancer and it would be more interesting if they find any difference between T cells in these two type of ICCs.

Reply: Thank you for your valuable suggestion. We first examine the differences of T cell subsets as well as exhausted CD8+ T cells (PD1+CD8+ T cells) and Treg cells between S100P+SPP1- iCCA^{phl} and S100P-SPP1+ iCCA^{pps} in our scRNA-seq data. As shown in Figure R12a, S100P+SPP1- iCCA^{phl} possessed higher proportions of PD1+CD8+ T cells than S100P-SPP1+ iCCA^{pps} ($P = 0.007$). Although there was no statistically significant difference in CD8+ and CD4+ T cells infiltration between these two subtypes in our scRNA-seq data, this difference was found in our staining of TMA cohort (68 S100P+SPP1- iCCA^{phl} and 118 S100P-SPP1+ iCCA^{pps}, Figure R12b, c). The results showed that iCCA^{phl} harbored increased CD8+ T cells and decreased CD4+ T cells than iCCA^{pps} ($P < 0.01$). Also, iCCA^{phl} displayed significantly higher PD1+CD8+ T cell infiltration than iCCA^{pps} ($P < 0.01$), while no significant difference was found in FOXP3+CD4+ Treg cells infiltration

(Figure R12d). The discrepancy in statistical results between scRNA-seq and TMA data partly due to the small sample size of scRNA-seq cases and the inability of mRNA level to thoroughly reflect protein expression. In summary, these results indicate that although S100P+SPP1- iCCA^{phl} has a higher proportion of CD8+ T cells, most of these cells are in a state of exhaustion. We have added these data to the revised manuscript (Page 16, Lines 336-340, and Supplementary Fig. 7a).

Figure R12 The differences of T cell subsets infiltrated in S100P+SPP1-

iCCA^{phl} and S100P-SPP1+ iCCA^{pps}. **a** Comparisons of the proportions of T cell subsets between S100P+SPP1- and S100P-SPP1+ iCCA in our scRNA-seq cohort. **b-d** Representative mIHC images (b, c) and statistical graphs (d) to show the distribution of T cell subsets in two iCCA subtypes. **P < 0.01. NS = not significant.

Reference

1. Hayashi A, *et al.* Distinct Clinicopathologic and Genetic Features of 2 Histologic Subtypes of Intrahepatic Cholangiocarcinoma. *Am J Surg Pathol* **40**, 1021-1030 (2016).
2. Komuta M, *et al.* Histological diversity in cholangiocellular carcinoma reflects the different cholangiocyte phenotypes. *Hepatology* **55**, 1876-1888 (2012).
3. Liao JY, Tsai JH, Yuan RH, Chang CN, Lee HJ, Jeng YM. Morphological subclassification of intrahepatic cholangiocarcinoma: etiological, clinicopathological, and molecular features. *Mod Pathol* **27**, 1163-1173 (2014).
4. Tsai JH, Huang WC, Kuo KT, Yuan RH, Chen YL, Jeng YM. S100P immunostaining identifies a subset of peripheral-type intrahepatic cholangiocarcinomas with morphological and molecular features similar to those of perihilar and extrahepatic cholangiocarcinomas. *Histopathology* **61**, 1106-1116 (2012).
5. Aishima S, *et al.* Different roles of S100P overexpression in intrahepatic cholangiocarcinoma: carcinogenesis of perihilar type and aggressive behavior of peripheral type. *Am J Surg Pathol* **35**, 590-598 (2011).
6. Radwan NA, Ahmed NS. The diagnostic value of arginase-1 immunostaining in differentiating hepatocellular carcinoma from metastatic carcinoma and cholangiocarcinoma as compared to HepPar-1. *Diagn Pathol* **7**, 149 (2012).
7. Ma L, *et al.* Single-cell atlas of tumor cell evolution in response to therapy in hepatocellular carcinoma and intrahepatic cholangiocarcinoma. *J Hepatol*, (2021).
8. Moorman HR, Poschel D, Klement JD, Lu C, Redd PS, Liu K. Osteopontin: A Key Regulator of Tumor Progression and Immunomodulation. *Cancers (Basel)* **12**, (2020).
9. Azizi E, *et al.* Single-Cell Map of Diverse Immune Phenotypes in the

- Breast Tumor Microenvironment. *Cell* **174**, 1293-1308 e1236 (2018).
10. Muller S, *et al.* Single-cell profiling of human gliomas reveals macrophage ontogeny as a basis for regional differences in macrophage activation in the tumor microenvironment. *Genome Biol* **18**, 234 (2017).
 11. Wang X, *et al.* Comparative analysis of cell lineage differentiation during hepatogenesis in humans and mice at the single-cell transcriptome level. *Cell Res* **30**, 1109-1126 (2020).
 12. Huang L, *et al.* ID3 Promotes Stem Cell Features and Predicts Chemotherapeutic Response of Intrahepatic Cholangiocarcinoma. *Hepatology* **69**, 1995-2012 (2019).
 13. Cheng S, *et al.* A pan-cancer single-cell transcriptional atlas of tumor infiltrating myeloid cells. *Cell* **184**, 792-809 e723 (2021).
 14. Zhang L, *et al.* Single-Cell Analyses Inform Mechanisms of Myeloid-Targeted Therapies in Colon Cancer. *Cell* **181**, 442-459 e429 (2020).
 15. Leader AM, *et al.* Single-cell analysis of human non-small cell lung cancer lesions refines tumor classification and patient stratification. *Cancer Cell*, (2021).
 16. Puram SV, *et al.* Single-Cell Transcriptomic Analysis of Primary and Metastatic Tumor Ecosystems in Head and Neck Cancer. *Cell* **171**, 1611-1624 e1624 (2017).
 17. Filbin MG, *et al.* Developmental and oncogenic programs in H3K27M gliomas dissected by single-cell RNA-seq. *Science* **360**, 331-335 (2018).
 18. Jerby-Arnon L, *et al.* A Cancer Cell Program Promotes T Cell Exclusion and Resistance to Checkpoint Blockade. *Cell* **175**, 984-997 e924 (2018).
 19. Tirosh I, *et al.* Dissecting the multicellular ecosystem of metastatic melanoma by single-cell RNA-seq. *Science* **352**, 189-196 (2016).
 20. Zou S, *et al.* Mutational landscape of intrahepatic cholangiocarcinoma. *Nat Commun* **5**, 5696 (2014).
 21. Nakamura H, *et al.* Genomic spectra of biliary tract cancer. *Nat Genet* **47**, 1003-1010 (2015).
 22. Jusakul A, *et al.* Whole-Genome and Epigenomic Landscapes of Etiologically Distinct Subtypes of Cholangiocarcinoma. *Cancer Discov* **7**, 1116-1135 (2017).
 23. Banales JM, *et al.* Cholangiocarcinoma 2020: the next horizon in mechanisms and management. *Nat Rev Gastroenterol Hepatol* **17**, 557-588 (2020).
 24. Asayama Y, *et al.* Coexpression of neural cell adhesion molecules and

bcl-2 in intrahepatic cholangiocarcinoma originated from viral hepatitis: relationship to atypical reactive bile ductule. *Pathol Int* **52**, 300-306 (2002).

25. Wu Y, *et al.* Spatiotemporal Immune Landscape of Colorectal Cancer Liver Metastasis at Single-Cell Level. *Cancer Discov*, (2021).
26. Inoue M, Shinohara ML. Intracellular osteopontin (iOPN) and immunity. *Immunol Res* **49**, 160-172 (2011).
27. Brunt E, *et al.* cHCC-CCA: Consensus terminology for primary liver carcinomas with both hepatocytic and cholangiocytic differentiation. *Hepatology* **68**, 113-126 (2018).
28. Xue R, *et al.* Genomic and Transcriptomic Profiling of Combined Hepatocellular and Intrahepatic Cholangiocarcinoma Reveals Distinct Molecular Subtypes. *Cancer Cell* **35**, 932-947 e938 (2019).

Reviewers' Comments:

Reviewer #2:

Remarks to the Author:

The authors have performed further experiments. They have also adequately addressed my comments and concerns.

Reviewer #3:

Remarks to the Author:

I have no further comments

Reviewer #4:

Remarks to the Author:

I am satisfied with their addresses to my comments. They did good job.